# GET THE GIST OF GRAPHS WITH INTERSECTION SIGNATURE

## ABSTRACT

Graph Transformers have emerged as a promising alternative to Graph Neural Networks (GNNs), offering global attention that mitigates oversmoothing and oversquashing issues. However, their success critically depends on how structural information is encoded, especially for graph-level tasks such as molecular property prediction. Existing positional and structural encodings capture some aspects of topology, yet overlook the diverse and interacting substructures that shape graph behavior. In this work, we introduce Gisty Intersection Signature Trait (GIST), a structural encoding based on the intersection cardinalities of k-hop neighborhoods between node pairs. GIST provides a permutation-invariant representation that is theoretically expressive, while remaining scalable through efficient randomized estimation. Incorporated as an attention feature, GIST enables Graph Transformers to capture fine-grained substructures together with node-pairwise relationships that underlie long-range interactions. Across diverse and comprehensive benchmarks, GIST maintains a uniformly strong performance profile: head-to-head evaluations consistently favor GIST, underscoring its role as a simple and expressive structural feature for Graph Transformers.

## 1 INTRODUCTION

Graph-level task is a foundational problem in machine learning with broad impact across chemistry, biology, and drug discovery (Dwivedi et al., 2022a;d; Irwin et al., 2012; Wu et al., 2017): It advances molecular property prediction, reveals complex biological interactions, and supports the discovery of new therapeutics. For these tasks, Graph Neural Networks (GNNs) (Kipf & Welling, 2017; Han et al., 2022) have been the primary choice, learning node- and graph-level representations via neighborhood aggregation. Yet their local message passing mechanism unfavorably carries well-know drawbacks including oversmoothing (Keriven, 2022), oversquashing (Black et al., 2023), and limited expressivity (Wang & Zhang, 2024).

Transformers (Vaswani et al., 2017) offer a compelling alternative for graph representation learning: global attention can connect distant nodes and model complex interactions, yielding strong performance on graph classification benchmarks (Ying et al., 2021). Nonetheless, adapting Transformers to graphs is nontrivial. Unlike sequential or image data, node indices exist but are arbitrary and carry no semantic meaning, so attention cannot rely on positional order or raw IDs to tell nodes apart. Without explicit structural priors, e.g., topology-aware positional/structural encodings or bias terms, the attention mechanism struggles to capture the complex relationships ubiquitous across all graphs.

In response, prior works have attempted to improve Transformers with graph structural inductive bias by integrating positional or structural features, such as shortest path distances (Ying et al., 2021), Laplacian eigenvector-based encodings (Kreuzer et al., 2021), and random walk-based features (Rampášek et al., 2022; Ma et al., 2023). While these methods provide some structural context, they either **fail to capture comprehensive substructural information** essential for distinguishing complex graph patterns (Rampášek et al., 2022) or **focus predominantly on a limited set of substructures** while **neglecting higher-order structural relationships** (Wollschlager et al., 2024). The challenge remains to identify a more expressive and comprehensive set of structural features and devise efficient methods for encoding them within the Transformer's self-attention mechanism.

In this work, we introduce Gisty Intersection Signature Trait (GIST), a novel structural feature characterizing the inherent substructures within a graph with $k$-hop node-pairwise neighborhood

intersections. Our approach is grounded in the theoretical understanding that the cardinality of the intersection between two nodes' $k$-hop neighborhoods can serve as an expressive permutation-invariant feature for substructure characterization. Used as a structural encoding, GIST enhances the Transformer's capability to comprehend complex graph patterns and their interactions. In contrast to prior works (Ma et al., 2023; Geisler et al., 2024; Rampášek et al., 2022) that focus on learning representations by aggregating similar substructures, GIST, to the best of our knowledge, is **the first to promote aggregation across heterogeneous substructures** by capturing higher-order relationships among them. We adopt an efficient randomized algorithm to estimate GIST, ensuring its scalability to a large (number of) graphs. Baseline-to-baseline comparisons across a comprehensive set of graph-level benchmarks consistently favor GIST, yielding non-marginal average gains and a uniformly strong performance profile.

Our key contributions are as follows:

- We introduce **GIST**, an expressive structural encoding based on pairwise $k$-hop *substructure vectors*, computed efficiently via randomized estimation of the *intersection cardinality* between the $k$-hop neighborhoods of node pairs.
- We incorporate GIST into the attention mechanism as a learnable structural feature, and provide both theoretical and empirical evidence for its expressiveness and effectiveness.
- We conduct comprehensive evaluations on standard graph-level benchmarks, observing consistently strong improvements over competitive baselines.

Taken together, GIST contributes to the advancement of structural encoding for Graph Transformers, enabling simpler yet more effective graph-level prediction.[1]

## 2 MOTIVATION

Transformers, originally designed for sequential data, lack an inherent mechanism to capture the structural biases of graph data as highlighted in Ying et al. (2021); Rampášek et al. (2022). Without a well-designed structural bias (structural encoding), they treat all nodes as equally related, failing to utilize the relational dependencies critical for graph tasks (Ying et al., 2021; Brody et al., 2022).

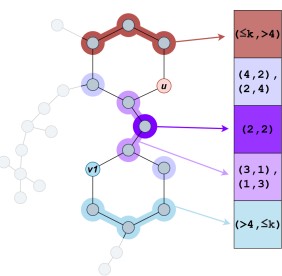
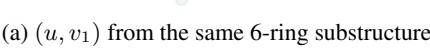
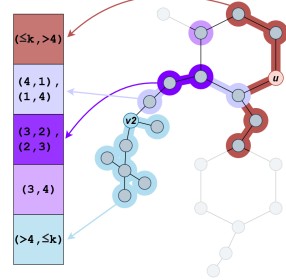

(a) $(u, v_1)$ from the same 6-ring substructure      (b) $(u, v_2)$ from different substructures

Figure 1: $k$-hop Substructure Vector Visualization (Def. 3.1) of ZINC molecule. The substructures of node pairs in the form of **intersection cardinality** of their common neighborhood at different distances from $u$ and $v$ are **"GIST"-ed** into the Substructure Vector. Specifically, each cell $(k_u, k_v)$ in the Substructure Vector denotes the number of nodes that are **exactly** $k_u$ hops from $u$ and $k_v$ hops from $v$. The variations in the Substructure Vector help the self-attention mechanism distinguish structural differences between node pairs, such as $(u, v_1)$ and $(u, v_2)$. For example, the pair $(u, v_1)$, which belongs to the **same** 6-ring substructure, has intersection cardinalities $\mathcal{I}_{(2,2)}(u, v_1) = \mathcal{I}_{(4,2)}(u, v_1) = \mathcal{I}_{(2,4)}(u, v_1) = 1$. In contrast, the pair $(u, v_2)$, where $u$ and $v_2$ belong to **different** substructures (a 6-ring and a 2-path), has $\mathcal{I}_{(2,2)}(u, v_2) = \mathcal{I}_{(4,2)}(u, v_2) = \mathcal{I}_{(2,4)}(u, v_2) = 0$.

**Challenge 1. Capturing Graph Substructures in Structural Encoding.** The first key challenge in designing effective structural encodings for Graph Transformers is capturing the substructures within a graph, as these substructures often represent critical local patterns, or fragments that define the

---

[1]The code will be made publicly available upon publication.

graph's overall characteristics (Ying et al., 2021; Ma et al., 2023; Wollschlager et al., 2024). While many early-stage structural encoding methods, such as shortest path distance (SPD) (Ying et al., 2021), provide a notion of proximity between nodes, they often struggle to effectively capture and represent substructures.

**Challenge 2. Aggregating Diverse Substructures Information.** As highlighted in Wollschlager et al. (2024), it is equally important for structural encodings to enable the aggregation of information across diverse substructures, rather than restricting it to similar or localized patterns. Graphs, such as molecules, often exhibit a variety of substructures that interact in complex ways, and limiting information flow to nodes in different structures can hinder the model's ability to capture global dependencies and cross-pattern interactions. This is particularly important in domains like chemistry, biology, and social networks, where functional or structural properties often arise from specific subgraph arrangements & interactions (i.e., rings and bonds in molecules) rather than the global graph structure alone (Yang et al., 2018; Yu & Gao, 2022). Many recent structural biases, such as shortest path distance (Ying et al., 2021) or those based on random walks (Rampášek et al., 2022; Ma et al., 2023), are effective at capturing simple substructures like cycles but tend to focus predominantly on these patterns, **neglecting the interactions between different substructures** (Wollschlager et al., 2024). For example, in Figure 2, it is more beneficial for $u$ to aggregate information from the 6-ring, X-shape, and 2-path substructures rather than solely focusing on another 6-ring that mirrors its own structural pattern. This highlights the need for a structural encoding that not only enables attention mechanisms to effectively learn substructural patterns, but also allows nodes to distinguish their own substructures from those of others, **guiding attention based on different structural relationships**.

**Observation 1: Intersection Cardinality as a Discriminative Subgraph Feature.** Empirically, we observe that the intersection cardinality of common neighborhoods between two nodes $(u, v)$ can serve as a powerful and discriminative feature encoding the $k-$hop subgraph structures. As illustrated in Figure 1, the intersections of common neighborhoods at different hop distances provide a structured way for $u$ to differentiate between the ring structure containing $v_1$ and the 2-path structure containing $v_2$, based on the differences in the in-between graph structures. Specifically, for $(u, v_1)$, which belongs to the same 6-ring substructure, the intersection cardinality values $\mathcal{I}_{(2,2)}$, $\mathcal{I}_{(4,2)}$, and $\mathcal{I}_{(2,4)}$ are all nonzero. In contrast, $(u, v_2)$, which belongs to different substructures (a 6-ring and a 2-path), lacks these intersection values but instead exhibits nonzero intersection cardinality in positions such as $\mathcal{I}_{(3,2)}$ and $\mathcal{I}_{(2,3)}$, which are absent for $(u, v_1)$. This contrast highlights how different substructure compositions lead to distinct intersection patterns, **enabling the model to effectively distinguish between structurally similar and dissimilar node pairs, guiding the self-attention mechanism based on higher-order relationships**.

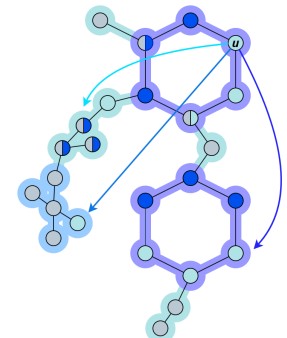

Figure 2: Node Clustering via Spectral Clustering Using Learned GIST Features in Graph Transformers on ZINC molecule graph. **Nodes within the same local substructures are clustered together**: 6-rings (purple), 2-path (cyan), and X-shape (light blue).

**Observation 2: Intersection Cardinality Enhances Structural Awareness in Self-Attention Mechanisms.** Moreover, our empirical results show that using intersection cardinality as an attention bias helps the attention mechanism effectively identify distinct substructures within the graph. In Figure 2, we train a Transformer architecture on the ZINC dataset (Dwivedi et al., 2022a), introducing only the intersection cardinality (formally defined in Section 4 as GIST) as a bias in the attention scores. After training the model, we apply Spectral Clustering to group nodes based on the learned GIST features. The GIST features facilitate representation aggregation across structurally similar regions, allowing node $u$ to integrate information from another ring structure. This effect is evident as nodes from both rings are grouped into the same clusters, marked in dark blue and cyan. Furthermore, certain nodes positioned at the boundaries of these substructures act as "information exchange points", facilitating communication between distant regions of the graph. For example, the cyan-colored node within the "X" substructure is assigned to the same cluster as the ring nodes, effectively **facilitating representation aggregation between two different substructures**—an ability that current GNNs and Graph Transformers often lack, as they tend to favor aggregation among structurally similar components. We note that this is **not a**

**cherry-picked example**; rather, this phenomenon **consistently occurs across multiple samples** of the trained Transformer on ZINC.

# 3 GIST: GISTY INTERSECTION SIGNATURE TRAIT

In this section, we formally introduce the Gisty Intersection Signature Trait (GIST). We begin with how to encode the $k$-hop substructure of a node pair $(u, v)$ based on the $k$-hop common neighborhood between them. Next, we introduce how to use encoded $k$-hop substructures in a graph to form GIST. Finally, we show how to efficiently compute GIST with randomized hashing algorithms.

**Notation:** We denote an undirected graph $\mathcal{G} = (\mathcal{V}, \mathcal{E})$, which contains a set $\mathcal{V}$ of $n$ nodes (vertices) and a set $\mathcal{E}$ of $m$ edges (links). Each node $v \in \mathcal{V}$ is associated with a $d_n$-dimensional feature $x_v \in \mathbb{R}^{d_n}$, while each edge $e_{u,v} \in \mathcal{E}$ connecting node pair $(u, v)$ is associated with a $d_e$-dimensional edge feature $y_{u,v} \in \mathbb{R}^{d_e}$ ($y_{u,v} = \mathbf{0}^{d_e}$ if there is no edge between $u$ and $v$). For every node $v \in \mathcal{V}$, we denote its $k$-hop neighborhoods as $\mathcal{N}_k(v)$: it consists of all the vertices whose shortest path distances from $v$ are less than or equal to $k$. Additionally, we define the $k$-hop common neighborhood of a node pair $(u, v)$ as $\mathcal{C}_{k_u, k_v}(u, v) = \mathcal{N}_{k_u}(u) \cap \mathcal{N}_{k_v}(v)$, which is the set of nodes in the graph that are within $k_u$-hop from $u$ and with $k_v$-hop from $v$, respectively.

## 3.1 ENCODING $k$-HOP SUBSTRUCTURE OF A NODE PAIR

Following (Chamberlain et al., 2022), we encode the $k$-hop substructure of a node pair $(u, v)$ by a vector. This vector is computed based on the $k$-hop common neighborhood $\mathcal{C}_{k_u, k_v}(u, v)$.

**Definition 3.1** ($k$-hop substructure vector)**.** Given a pair of nodes $(u, v) \in \mathcal{G}$, we propose to capture the $k$-hop graph structure between $u$ and $v$ with two types of features computed by $k$-hop common neighborhood $\mathcal{C}_{k_u, k_v}(u, v)$ as follows. For all $1 \leq k_u, k_v \leq k$, let

- $\mathcal{I}_{k_u, k_v}(u, v)$ (internal node counts): the cardinality of common neighborhoods that are exactly $k_u$ hops from node $u$ and $k_v$ hops from node $v$, computed as:

$$\mathcal{I}_{k_u, k_v}(u, v) = |\mathcal{C}_{k_u, k_v}(u, v)| - \sum_{\substack{1 \leq x \leq k_u \ , \ 1 \leq y \leq k_v \\ (x,y) \neq (k_u, k_v)}} \mathcal{I}_{x,y}(u, v),$$

where $\mathcal{I}_{1,1}(u, v) = |\mathcal{C}_{1,1}(u, v)|$ for $u$ and $v$.

- $\mathcal{B}_{k_u, >k}(u, v)$ (boundary node counts): the cardinality of nodes that are exactly $k_u$ hop from vertex $u$ and greater than $k$ hop from $v$ (and vice-versa for $\mathcal{B}_{k_v, >k}(v, u)$), computed as:

$$\mathcal{B}_{k_u, >k}(u, v) = |\mathcal{N}_{k_u}(u)| - \sum_{k_v=1}^{k} \mathcal{I}_{k_u, k_v}(u, v)$$

For any ordered node pair $(u, v)$, there are $k^2$ entries of $\mathcal{I}_{k_u, k_v}(u, v)$, $k$ entries of $\mathcal{B}_{k_u, >k}(u, v)$, and $k$ entries of $\mathcal{B}_{k_v, >k}(v, u)$. Finally, we encode the $k$-hop graph substructure for every ordered node pair $(u, v)$ as a $(k^2 + 2k)$-dimensional vector[2] $S_k(u, v)$: the first $k^2$ components of $S_k(u, v)$ are entries of $\mathcal{I}_{k_u, k_v}(u, v)$ for every pair of $1 \leq k_u, k_v \leq k$; we then fill the remaining dimension in $S_k(u, v)$ with $\mathcal{B}_{k_u, >k}(u, v)$ for each $k_u \leq k$ hop and $\mathcal{B}_{k_v, >k}(v, u)$ for each $k_v \leq k$ hop.

## 3.2 GIST: GISTY INTERSECTION SIGNATURE TRAIT

With $k$-hop substructure encoding $S_k(u, v)$ for every ordered node pair $(u, v) \in \mathcal{V} \times \mathcal{V}$, we define our new *Gisty Intersection Signature Trait* (GIST) encoding of every node $u \in \mathcal{V}$ and subsequently the entire graph $\mathcal{G}$.

**Definition 3.2** (Gisty Intersection Signature Trait (GIST))**.** Let $\mathcal{G} = (\mathcal{V}, \mathcal{E})$ be a graph with $n$ nodes ($|\mathcal{V}| = n$), and $k > 0$ be an integer. For any ordered node pair $(u, v) \in \mathcal{V} \times \mathcal{V}$, let $S_k(u, v)$ be the

---

[2]If we add an additional entry $\mathcal{B}_{>k,>k}(u, v) = \perp$ (or any other special symbol), then one may view these $k^2$ entries of $\mathcal{I}_{k_u, k_v}(u, v)$, $2k$ entries of $\mathcal{B}_{k_u, >k}(u, v)$ and $\mathcal{B}_{k_v, >k}(v, u)$, together with the extra entry, form a $(k+1) \times (k+1)$ *distance matrix*. Then $S_k(u, v)$ is just a vectorization of this distance matrix with the extra entry removed.

$k$-hop graph substructure encoding of $(u, v)$ defined in Definition 3.1. Then the GIST feature vector (or coloring) of any node $u \in \mathcal{V}$ is defined as

$$\chi_u = \text{hash}\left(\{\{S_k(u, v) : v \in \mathcal{V}\}\}\right),$$

where $\{\{...\}\}$ denotes a multiset.

The GIST encoding of graph $G$ is then defined by $\chi(G) = \{\{\chi_u : u \in \mathcal{V}\}\}$.

In fact, one may alternatively view the GIST encoding of a graph $\mathcal{G} = (\mathcal{V}, \mathcal{E})$ as a three-dimensional tensor $x(G) \in \mathbb{R}^{n \times n \times (k^2 + 2k)}$. A fixed-length representation of each multiset $S_k(u, v)$ is obtained by imposing a consistent ordering (e.g., lexicographic) on its elements; if in addition a length-preserving hash function is applied to compute node feature vectors, then every $x_u$ is a matrix of dimension $n \times (k^2 + 2k)$. It follows that the encoding of $G$, $x(G)$, is a 3-tensor of dimension $n \times n \times (k^2 + 2k)$.

Unlike Chamberlain et al. (2022) sketching the subgraph between a node pair, GIST provides a compact representation of a graph's structural properties, encoding its topology and connectivity patterns by capturing higher-order relational dependencies among nodes and substructures. This encoding enables the differentiation of substructures, offering a detailed understanding of complex higher-order relationships, as illustrated in Figure 2 and Section 2. We would like to note one component of this representation: the diagonal entry $S_k(u, u)$, which essentially encodes the $k$-hop neighborhood surrounding a node $u \in \mathcal{V}$. This local structure provides a positional reference that differentiates nodes based on their placement within the global graph topology, enabling the model to capture long-range dependencies beyond direct connectivity.

### 3.3 ON THE EXPRESSIVENESS OF GIST: A THEORETICAL PERSPECTIVE

We now compare the expressive power of GIST with some other popular graph invariants. In the following, we use GIST($k$) to denote our GIST encoding with hop-neighborhood radius $k$.

Recently, Zhang et al. (2023b) proposed the *Generalized Distance Weisfeiler-Leman Test* (GD-WL) — a graph isomorphism test based on incorporating the distances between a node with all other nodes in the graph into the encoding of that node. Let $\mathcal{G} = (\mathcal{V}, \mathcal{E})$ be a graph, $d(u, v)$ denotes a distance between nodes $u$ and $v$. Then the GD-WL encoding of a node $u \in \mathcal{V}$ is defined as

$$\chi(u) = \text{hash}\left(\chi_0(u), \{\{d(u, v) : v \in \mathcal{V}\}\}\right),$$

where $\chi_0(u)$ denotes the initial coloring of vertex $u$. Zhang et al. (2023b) analyze a Graph Transformer architecture that uses $d(u, v)$ as a relative positional encoding. They show that choosing $d(u, v)$ as the shortest-path distance $d^{\text{SPD}}(u, v)$ allows the model to solve edge biconnectivity. This corresponds to the Shortest-Path-Distance Weisfeiler–Leman variant (SPD-WL). Likewise, using the resistance distance $d^{\text{RD}}(u, v)$ enables the model to solve vertex biconnectivity. This corresponds to the Resistance-Distance Weisfeiler–Leman variant (RD-WL).

Let $A \in \{0, 1\}^{n \times n}$ be the adjacency matrix of a graph $\mathcal{G} = (\mathcal{V}, \mathcal{E})$ with $n$ nodes, and let $D$ be the diagonal degree matrix, i.e. $D_{u,v} = \delta(u, v) \sum_{x \in \mathcal{V}} A(u, x)$, where $\delta(u, v)$ is the Kronecker delta function. Define $M = D^{-1} A$, and note that $M_{u,v}$ is the probability that $u$ hops to $v$ in one step of a simple random walk. More generally, $M_{u,v}^k$ is the probability that a simple random walk of length $k$ starting from node $u$ ends at node $v$. Let $k$ be a fixed positive integer, then for each pair of nodes $(u, v) \in \mathcal{V} \times \mathcal{V}$, define:

$$P_{u,v}^k = \left(I_{u,v}, M_{u,v}, M_{u,v}^2, \ldots, M_{u,v}^{k-1}\right)$$

where $I$ is the identity matrix. The so-called *relative random walk probabilities* (RRWP($k$)) positional encoding, extensively studied in e.g. Dwivedi et al. (2022c); Ma et al. (2023), is defined by, for every $u \in \mathcal{V}$,

$$\chi(u) = \text{hash}\left(\chi_0(u), \{\{P_{u,v}^k : v \in \mathcal{V}\}\}\right),$$

where, once again, $\chi_0(u)$ denotes the initial coloring of vertex $u$.

**Theorem 3.3.** *For the expressive power of GIST, we have the following:*

- *GIST($n - 1$) is more expressive than SPD-WL.*

- *There exists a pair of graphs such that* GIST($n - 1$) *distinguishes them while RD does not.*

- *There exist a pair of graphs such that* GIST *distinguishes them while* RRWP *does not.*

The proof of Theorem 3.3 as well as definitions of related concepts can be found in Appendix C.

Table 1: Performance on GNNBenchmark datasets and ZINC-full.

| Model | ZINC-full (MAE $\downarrow$) | ZINC (MAE $\downarrow$) | MNIST (Accuracy $\uparrow$) | CIFAR10 (Accuracy $\uparrow$) |
|---|---|---|---|---|
| GCN (Kipf & Welling, 2017) | $0.113 \pm 0.002$ | $0.367 \pm 0.011$ | $0.907 \pm 0.002$ | $0.557 \pm 0.004$ |
| GIN (Xu et al., 2018) | $0.088 \pm 0.002$ | $0.526 \pm 0.051$ | $0.965 \pm 0.003$ | $0.553 \pm 0.015$ |
| DS-GNN (Bevilacqua et al., 2023) | - | $0.087 \pm 0.003$ | - | - |
| GNN-SSWL (Zhang et al., 2023a) | $0.026 \pm 0.001$ | $0.082 \pm 0.003$ | - | - |
| GNN-SSWL+ (Zhang et al., 2023a) | $0.022 \pm 0.001$ | $0.070 \pm 0.005$ | - | - |
| GatedGCN-LSPE (Dwivedi et al., 2022d) | - | $0.090 \pm 0.001$ | $0.973 \pm 0.001$ | $0.673 \pm 0.003$ |
| Subgraphormer (Bar-Shalom et al., 2024) | $0.023 \pm 0.001$ | $0.063 \pm 0.001$ | - | - |
| FragNet (Wollschläger et al., 2024) | $0.024$ | $0.078 \pm 0.005$ | - | - |
| GRIT (Ma et al., 2023) | $0.023 \pm 0.001$ | $0.059 \pm 0.002$ | $0.981 \pm 0.001$ | $0.765 \pm 0.009$ |
| GraphGPS (Rampášek et al., 2022) | - | $0.070 \pm 0.004$ | $0.980 \pm 0.001$ | $0.723 \pm 0.004$ |
| TIGT (Choi et al., 2024) | $0.014 \pm 0.001$ | $0.057 \pm 0.002$ | $0.982 \pm 0.001$ | $0.739 \pm 0.004$ |
| SPSE (Airale et al., 2025) | - | $0.059 \pm 0.002$ | $0.983 \pm 0.001$ | $0.770 \pm 0.004$ |
| CSA (Menegaux et al., 2024) | - | $0.056 \pm 0.002$ | - | - |
| Graphormer (Kreuzer et al., 2021) | $0.052 \pm 0.005$ | $0.122 \pm 0.006$ | - | - |
| Graphormer-GD (Kreuzer et al., 2021) | $0.025 \pm 0.004$ | $0.081 \pm 0.009$ | - | - |
| GIST (ours) | $0.019 \pm 0.002$ | $0.050 \pm 0.002$ | $0.990 \pm 0.001$ | $0.781 \pm 0.003$ |

## 3.4 EFFICIENTLY COMPUTE GIST WITH RANDOMIZED HASHING

In this section, we show how to efficiently compute GIST by reducing the time complexity from $\mathcal{O}(k^2 n^4)$ to $\mathcal{O}(k^2 n^2)$. It is not hard to see that computing GIST $S(\mathcal{G})$ can be done in $\mathcal{O}(k^2 n^4)$ time. Indeed, note that for a node pair $(u, v)$, the exact computation of their $k$-hop common neighborhood $\mathcal{C}_{k_u, k_v}(u, v)$ incurs a cost of $\mathcal{O}(n^2)$, while calculating $S_{u,v}(\mathcal{G})$ requires $\mathcal{O}(k^2 n^2)$. Consequently, computing $S_{u,v}(\mathcal{G})$ for $n^2$ node pairs in a graph $\mathcal{G}$ results in an overall complexity of $\mathcal{O}(k^2 n^4)$. Exact intersection calculations are computationally expensive, making them impractical for large graphs. Adopting methods in Chamberlain et al. (2022); Le et al. (2024), we efficiently and unbiasedly estimate the cardinality of $k$-hop common neighborhood $\mathcal{C}_{k_u, k_v}(u, v)$ by decomposing it as:

$$|\mathcal{C}_{k_u, k_v}(u, v)| = \mathcal{J}_{k_u, k_v}(u, v) \cdot \mathcal{U}_{k_u, k_v}(u, v) \tag{1}$$

Here, $\mathcal{J}_{k_u, k_v}(u, v)$ represents the Jaccard similarity between $k_u$-hop neighborhoods $\mathcal{N}_{k_u}(u)$ and $k_v$-hop neighborhoods $\mathcal{N}_{k_v}(v)$. $\mathcal{U}_{k_u, k_v}(u, v)$ denotes the cardinality of the union $\mathcal{N}_{k_u}(u) \cup \mathcal{N}_{k_v}(v)$. Next, we can estimate $\mathcal{J}_{k_u, k_v}(u, v)$ with the constant-time collisions of the MinHash signatures of $\mathcal{N}_{k_u}(u)$ and $\mathcal{N}_{k_v}(v)$. We note that MinHash provides an unbiased estimator to the $\mathcal{J}_{k_u, k_v}(u, v)$ since the collision probability between the MinHash signatures of $\mathcal{N}_{k_u}(u)$ and $\mathcal{N}_{k_v}$ are equal to $\mathcal{J}_{k_u, k_v}(u, v)$ We can also estimate $\mathcal{U}_{k_u, k_v}(u, v)$ with the mergeable HyperLogLog signatures. We note that HyperLogLog also provides an unbiased estimator to $\mathcal{U}_{k_u, k_v}(u, v)$.

Finally, we multiply the estimated $\tilde{\mathcal{J}}_{k_u, k_v}(u, v)$ and $\tilde{\mathcal{U}}_{k_u, k_v}(u, v)$ together and form an unbiased estimator to $|\mathcal{C}_{k_u, k_v}(u, v)|$. This unbiased estimation can serve as an efficient alternative to exact computation for $|\mathcal{C}_{k_u, k_v}(u, v)|$. With MinHash and HyperLogLog, we reduce the computation time for $S_{u,v}(\mathcal{G})$ from $\mathcal{O}(k^2 n^2)$ to $\mathcal{O}(k^2)$, leading to $\mathcal{O}(k^2 n^2)$ time for GIST computation(see Appendix D for the detailed randomized algorithms used for these estimations in constant time)

## 4 GRAPH TRANSFORMERS GET THE GIST

We now show that GIST can be naturally integrated into Graph Transformers for graph structural encoding in the self-attention mechanism. As a result, we introduce the GIST attention for graph transformers.

**Definition 4.1.** Let $\mathcal{G} = (\mathcal{V}, \mathcal{E})$ denote a graph with $n$ nodes ($|\mathcal{V}| = n$). We view a node representation (or coloring) of graphs as a map $\chi : G \mapsto \chi_G$, such that $\chi_G : V(G) \to \mathcal{C}$ assign every vertex $v$ of $G$ a color $\chi_G(v)$ from the set of colors $\mathcal{C}$. A node representation (or coloring) is said to be *isomorphism invariant* if for any pair of isomorphic graphs $G$ and $H$ with $f$ being any isomorphism from $G$ to $H$, we have $\chi_H(f(v)) = \chi_G(v)$ for all vertex $v$ of $G$. Similarly, an edge representation (or coloring) $\chi_G : V(G) \times V(G) \to \mathcal{C}$ is said to be *isomorphism invariant* if for every isomorphism $f$ from a graph $G$ to a graph $H$, we have $\chi_H(f(u), f(v)) = \chi_G(u, v)$ for every edge $(u, v)$ in $G$.

**Definition 4.2** (GIST attention). Let $\mathcal{G} = (\mathcal{V}, \mathcal{E})$ denote a graph with $n$ nodes ($|\mathcal{V}| = n$). Let $x_u \in \mathbb{R}^{d_n}$ denote some initial isomorphism invariant representation of node $u \in \mathcal{V}$. Let $y_{u,v} \in \mathbb{R}^{d_e}$ denote some initial isomorphism invariant representation of edge between nodes $u, v \in \mathcal{V}$. Let $w_v \in \mathbb{R}^{d_n \times d_n}$ and $w_e \in \mathbb{R}^{d_n \times d}$ denote the model weight. Let $S_k(u, v)$ denote the $k$-hop GIST encoding computed from $\mathcal{G}$ (see Definition 3.1). We define the GIST attention as a transform $\psi : \mathbb{R}^{d_n} \to \mathbb{R}^{d_n}$ on every node feature $x_u$ as:

$$\psi(x_u) = \sum_{v \in \mathcal{V}} \mathcal{A}_{u,v} \cdot (w_v x_v + w_e \hat{\mathcal{A}}_{u,v}).$$

Here $\hat{\mathcal{A}}_{u,v} \in \mathbb{R}^d$ and attention score $\mathcal{A}_{u,v} \in \mathbb{R}$ are computed as follows:

$$e_{u,v} = \phi_y(y_{u,v}) + \phi_S(S_k(u, v))$$
$$\mathcal{A}_{u,v} = \sigma\big(\langle w_Q x_u + w_K x_v + w_b, e_{u,v}\rangle\big), \quad \hat{\mathcal{A}}_{u,v} = (w_Q x_u + w_K x_v + w_b) \odot e_{u,v},$$

where $\phi_y : \mathbb{R}^{d_e} \to \mathbb{R}^d$ and $\phi_S : \mathbb{R}^{k^2+2k} \to \mathbb{R}^d$ are MLP networks that align the representations of edge and GIST (see Definition 3.2) into vectors of the same dimension $d$ for addition, and $w_Q, w_K \in \mathbb{R}^{d \times d_n}$ and $w_b \in \mathbb{R}^d$ are model weights and bias, respectively. $\sigma$ is any non-linear activation function.

## 5 EXPERIMENT

We rigorously evaluate the effectiveness of GIST by addressing the following key research questions and providing corresponding insights:

- **RQ 1**: How strong and consistent is the performance on graph representation learning of Graph Transformer with GIST as a structural encoding?

- **RQ 2**: To what extent does GIST enable long-range dependencies in Graph Transformers?

- **RQ 3**: How sensitive is GIST to different hyperparameter settings?

- **RQ 4**: How well does GIST generalize to beyond graph-level task?

### 5.1 SETTINGS

We evaluate the proposed method on three benchmark suites comprising a total of 14 datasets, spanning small-scale to large-scale settings: the Long-Range Graph Benchmark (LRGB) (Dwivedi et al., 2022d), MoleculeNet (Wu et al., 2017), GNNBenchmark Dataset (Dwivedi et al., 2022a), and ZINC-full (Irwin et al., 2012). These datasets are curated to emphasize challenges in structural encoding and long-range dependency modeling, with diverse applications in domains such as chemistry.

**Baselines.** We benchmark the performance of our method against recent state-of-the-art baselines, including Graph Transformers, GNNs and hybrid models, as well as pretrained graph models: GraphGPS (Rampášek et al., 2022), GRIT (Ma et al., 2023), Subgraphormer (Bar-Shalom et al., 2024), FragNet (Wollschlager et al., 2024), TIGT (Choi et al., 2024), SPSE (Airale et al., 2025), CSA (Menegaux et al., 2024), GatedGCN (Dwivedi et al., 2022d), SAN (Kreuzer et al., 2021), Graphormer (Ying et al., 2021), Graphormer-GD (Zhang et al., 2023b), GCN (Kipf & Welling, 2017), GIN (Xu et al., 2018), DS-GNN (Bevilacqua et al., 2022), DSS-GNN (Bevilacqua et al., 2022), GNN-SSWL(Zhang et al., 2023a), GraphMVP (Liu et al., 2022), MGSSL (Zhang et al., 2021), and GraphFP (Luong & Singh, 2023).

**Experimental Settings.** For each dataset, we train our method on the training split and select the epoch that achieves the best validation performance. The corresponding test results are then reported. All results for our method (and baselines reproduction) are averaged over five runs with different random seeds and presented as mean $\pm$ standard deviation. Baseline performance is taken from original publications when available or reproduced using their reported best hyperparameters. Top-3 Results Highlighted in **Red**, **Blue**, and **Orange**. (see Appendix E for details)

**Hyperparameters.** Particularly for our method, we perform a grid search to find the optimal hyperparameter combination for each dataset whenever feasible. The intersection features are within [1,2,3,4,5,6]-hops of each node, the batch size is chosen among [32, 64, 128, 256], the number of

layers is chosen among [2, 4, 6, 8], the number of heads is chosen among [2, 4, 8, 16, 32], the number of hidden dimensions is chosen among [16, 32, 64, 128], and learning rate is chosen among [0.0001, 0.0003, 0.0005, 0.002]. The chosen optimizer is AdamW. Our model is trained at 200 epochs for all datasets, except for MUV and HIV, where it is trained for 100 epochs. All model training and evaluations were conducted on NVIDIA A100 GPUs with 80G memory. Appendix E provides additional details on the experimental settings, including dataset statistics.

## 5.2 LONG-RANGE GRAPH BENCHMARK (LRGB)

We evaluate the ability of our proposed GIST to learn long-range dependencies using two graph classification datasets from LRGB (Dwivedi et al., 2022d): Peptides-func and Peptides-struct. These datasets provide a robust benchmark for assessing graph classification methods in handling long-range dependencies and addressing structural challenges such as over-squashing and over-smoothing of many GNNs. As shown in Table 2, GIST significantly enhances the capability of Transformers, achieving strong performance on

Table 2: Performance of GIST on Peptides datasets.

| Model | Peptides-struct MAE ↓ | Peptides-func AP ↑ |
|---|---|---|
| GCN (Kipf & Welling, 2017) | 0.2460± 0.0007 | 0.6860 ± 0.0050 |
| GIN (Xu et al., 2018) | 0.3547 ± 0.0045 | 0.5498 ± 0.0079 |
| Subgraphormer (Bar-Shalom et al., 2024) | 0.2494 ± 0.0020 | 0.6415 ± 0.052 |
| FragNet (Wollschlager et al., 2024) | 0.2462 ± 0.0021 | 0.6678 ± 0.0050 |
| GatedGCN+RWSE (Dwivedi et al., 2022d) | 0.2477 ± 0.0009 | 0.6765 ± 0.0047 |
| GRIT (Ma et al., 2023) | 0.2460± 0.0012 | 0.6988± 0.0082 |
| GraphGPS (Rampášek et al., 2022) | 0.2509 ± 0.0012 | 0.6534 ± 0.0041 |
| TIGT (Choi et al., 2024) | 0.2485 ± 0.0015 | 0.6679 ± 0.0074 |
| SPSE (Airale et al., 2025) | 0.2449± 0.0018 | 0.6945 ± 0.0113 |
| SAN+LapPE (Kreuzer et al., 2021) | 0.2683 ± 0.0043 | 0.6384 ± 0.0121 |
| SAN+RWSE (Kreuzer et al., 2021) | 0.2545 ± 0.0012 | 0.6439 ± 0.0075 |
| GNN-SSWL+ (Zhang et al., 2023a) | 0.2570 ± 0.006 | 0.5847 ± 0.0050 |
| GIST (ours) | 0.2442 ± 0.0011 | 0.6983 ± 0.0087 |

Peptides-struct while maintaining competitive result against recent SOTA baselines. Regarding **RQ2**, our results demonstrate that GIST effectively captures long-range dependencies by encoding structural relationships beyond local neighborhoods, leading to improved long-range graph-level task performance.

## 5.3 GNNBENCHMARK AND ZINC-FULL

We evaluate GIST on two molecular property prediction benchmarks(ZINC (Dwivedi et al., 2022a) & ZINC-full (Irwin et al., 2012)) and two graph classification datasets (MNIST & CIFAR10) from Dwivedi et al. (2022a). ZINC datasets are widely used to assess a model's ability to learn chemically meaningful representations from molecular graphs. ZINC features constrained molecular structures and well-defined tasks, making it a standard testbed for evaluating how well models capture local substructures associated with specific chemical properties. ZINC-full extends this to a larger and more diverse chemical space, testing generalization across broader molecular variations. As shown in Table 1, GIST significantly improves Transformer performance by enabling more effective modeling of chemically relevant substructures and their complex interaction.

## 5.4 MOLECULENET BENCHMARK

To further assess the effectiveness of GIST in molecular representation learning, we evaluate it on the MoleculeNet benchmark (Wu et al., 2017), a comprehensive suite of molecular property prediction tasks. MoleculeNet covers diverse real-world applications—ranging from drug discovery to toxicity prediction. As shown in Table 3, GIST consistently outperforms, or matches, state-of-the-art pre-trained graph models and Graph Transformers across multiple tasks.

Table 3: Performance on MoleculeNet: Top-3 Results Highlighted in **Red**, **Blue**, and **Orange**.

| Model | BBBP | Tox21 | Toxcast | Sider | Clintox | Bace | MUV | HIV | Avg. AUC |
|---|---|---|---|---|---|---|---|---|---|
| AttrMasking (Hu et al., 2020a) | 64.3 ± 2.8 | 76.7 ± 0.4 | 64.2 ± 0.5 | 61.0 ± 0.7 | 71.8 ± 4.1 | 79.3 ± 1.6 | 74.7 ± 1.4 | 77.2 ± 1.1 | 71.2 |
| GRIT (Ma et al., 2023) | 69.9 ± 1.3 | 75.9 ± 0.6 | 65.6 ± 0.4 | 60.3 ± 1.2 | 85.9 ± 2.9 | 84.4 ± 1.2 | 77.1 ± 1.7 | 77.3 ± 1.5 | 74.8 |
| GraphGPS (Rampášek et al., 2022) | 56.2 ± 4.4 | 71.4 ± 0.7 | 60.6 ± 1.0 | 60.2 ± 1.1 | 79.2 ± 3.6 | 71.5 ± 6.0 | 65.2 ± 1.6 | 66.0 ± 9.4 | 66.3 |
| GraphLoG (Xu et al., 2021) | 67.8 ± 1.9 | 75.1 ± 1.0 | 62.4 ± 0.2 | 59.5 ± 1.5 | 65.3 ± 3.2 | 80.2 ± 3.5 | 73.6 ± 1.2 | 73.7 ± 0.9 | 69.7 |
| GraphCL (You et al., 2020) | 69.7 ± 0.7 | 73.9 ± 0.7 | 62.4 ± 0.6 | 60.5 ± 0.9 | 76.0 ± 2.7 | 75.4 ± 1.4 | 69.8 ± 2.7 | 78.5 ± 1.2 | 70.8 |
| G-Motif (Rong et al., 2020) | 66.9 ± 3.1 | 73.6 ± 0.7 | 62.3 ± 0.6 | 61.0 ± 1.5 | 77.7 ± 2.7 | 73.0 ± 3.3 | 73.0 ± 1.8 | 73.8 ± 1.2 | 70.2 |
| G-Contextual (Rong et al., 2020) | 69.2 ± 3.0 | 75.0 ± 0.6 | 62.8 ± 0.7 | 58.7 ± 1.0 | 60.6 ± 5.2 | 79.3 ± 1.1 | 72.1 ± 0.7 | 76.3 ± 1.5 | 69.3 |
| GPT-GNN (Hu et al., 2020b) | 64.5 ± 1.4 | 74.9 ± 0.3 | 62.5 ± 0.4 | 58.1 ± 0.3 | 58.3 ± 5.2 | 77.9 ± 3.2 | 75.9 ± 2.3 | 65.2 ± 2.1 | 67.2 |
| GraphFP (Luong & Singh, 2023) | 72.0 ± 1.7 | 74.0 ± 0.7 | 63.9 ± 0.9 | 63.6 ± 1.2 | 84.7 ± 5.8 | 80.5 ± 1.8 | 75.4 ± 1.9 | 78.0 ± 1.5 | 74.0 |
| MGSSL (Zhang et al., 2021) | 68.9 ± 2.5 | 74.9 ± 0.6 | 63.3 ± 0.5 | 57.7 ± 0.7 | 67.5 ± 5.5 | 82.1 ± 2.7 | 73.2 ± 1.9 | 75.7 ± 1.3 | 70.4 |
| GraphMVP (Liu et al., 2022) | 68.5 ± 0.2 | 74.5 ± 0.4 | 62.7 ± 0.1 | 62.3 ± 1.6 | 79.0 ± 2.5 | 76.8 ± 1.1 | 75.0 ± 1.4 | 74.8 ± 1.4 | 71.7 |
| GIST (ours) | 73.6 ± 1.8 | 77.2 ± 0.4 | 67.3 ± 0.9 | 61.3 ± 2.7 | 88.2 ± 2.2 | 86.0 ± 1.9 | 75.5 ± 3.2 | 77.0 ± 0.2 | 75.8 |

## 5.5 ABLATION STUDY ON HYPERPARAMETERS

In order to analyze the impact of different hyperparameter settings on GIST, we conduct an ablation study on three key components: the number of $k$-hops, the number of MinHash functions, and the $p$ parameter of the HyperLogLog data structure. These experiments are performed across three datasets: ZINC, Peptides-struct, and Peptides-func. The $k$ value determines the extent of local versus long-range structural information captured, while the number of MinHash functions and the HyperLogLog $p$ parameter control the error of GIST's randomized cardinality estimation.

Table 4: Ablation study on different values of *k-hops*

| *k-hops* | 1 | 2 | 3 | 4 | 5 |
|---|---|---|---|---|---|
| ZINC | 0.100 | 0.058 | 0.050 | 0.065 | 0.063 |
| Peptides-struct | 0.2832 | 0.2471 | 0.2442 | 0.2478 | 0.2518 |
| Peptides-func | 0.6446 | 0.6420 | 0.6790 | 0.6754 | 0.6953 |

As shown in Table 4, GIST exhibits strong robustness to variations in the maximum hop distance $k$. Performance improves as $k$ increases from 1 to 3, reflecting GIST's ability to capture richer structural dependencies. Beyond $k = 3$, the changes in performance are minimal, and any decline is marginal, suggesting that GIST balances local expressiveness and global aggregation effectively without being overly sensitive to neighborhood size.

Table 5: Ablation study on different values of *MinHash functions*

| *# MinHash Functions* | 32 | 64 | 128 | 256 |
|---|---|---|---|---|
| ZINC | 0.071 | 0.069 | 0.069 | 0.049 |
| Peptides-struct | 0.2511 | 0.2538 | 0.2442 | 0.2444 |
| Peptides-func | 0.6502 | 0.6418 | 0.6519 | 0.6987 |

Table 5 examines the effect of varying the number of MinHash functions. While fewer hash functions (e.g., 32 or 64) can lead to slight variability in performance, increasing the number to 128 or 256 provides more stable and accurate intersection cardinality estimation. Notably, GIST performs well across a wide range of values, indicating tolerance to different trade-offs between estimation accuracy and computational overhead. Similarly, Table 6 shows that GIST is robust to different values of the HyperLogLog precision parameter $p$. While increasing $p$ generally improves cardinality estimation, the performance gains are modest, and all tested values yield competitive results. This suggests that GIST's randomized estimation pipeline remains reliable even under coarse-grained settings, enabling efficient scaling without sacrificing accuracy, answering **RQ3**.

Table 6: Ablation study on HyperLogLog data structurs with different values of *p*

| *p* | 4 | 6 | 8 | 10 |
|---|---|---|---|---|
| ZINC | 0.065 | 0.065 | 0.049 | 0.062 |
| Peptides-struct | 0.2566 | 0.2545 | 0.2442 | 0.2466 |
| Peptides-func | 0.6170 | 0.6124 | 0.6957 | 0.6771 |

## 5.6 GIST'S GENERALIZATION AND SCALABILITY

While GIST is primarily developed for graph-level tasks, we demonstrate its strong generalization and scalability across a broader range of settings, answering **RQ4**. We evaluate GIST on two node-level prediction benchmarks—Pattern and Cluster (Dwivedi et al., 2022a)—as well as the large-scale graph regression dataset PCQM4Mv2 (Hu et al., 2021). As shown in Table 13, GIST maintains competitive or even superior performance across all three tasks. These results suggest that its ability to model meaningful substructures and their higher-order interactions remains effective across varying scenarios. In addition, GIST scales efficiently to large graphs, benefiting from its efficient randomized estimation algorithm (Sec. 3.4).

Regarding **RQ1**, we conduct comparative evaluations against a broad set of methods across a comprehensive suite of graph-level benchmarks, so close runners inevitably arise in each individual settings. However, in pairwise comparisons carried out across all datasets, **no baseline demonstrates a comparably consistent level of strong performance to GIST** (see Table 14 and Table 15). The observed gaps indicate non-marginal, general-scale gains, reflecting that GIST delivers strong and competitive performance across diverse benchmarks.

# 6    RELATED WORKS

Recent work in graph representation learning emphasizes substructure modeling and Transformer-based architectures. Traditional GNNs struggle with complex structures due to over-smoothing and over-squashing. Alternatives like motif-based models, WL kernels, and spectral features improve expressiveness but face scalability or adaptability issues. Graph Transformers address these limits using self-attention, positional encodings, and structure-aware mechanisms to better capture graph topology.

Chamberlain et al. (2022) propose using MinHash and HyperLogLog to sketch local node-centric subgraphs for the purpose of resolving the automorphic node problem in link prediction. Although we adopt the same estimators for efficiency, GIST targets a fundamentally different structural quantity: $k$-hop intersection patterns between all node pairs. GIST is designed as a global structural feature for Graph Transformers. As such, GIST captures higher-order relationships among heterogeneous substructures rather than summaries of local subgraphs, which differentiates both its intent and its use within attention. We refer the readers to Appendix B for a detailed discussion of related works.

# 7    CONCLUSION

This paper presents Gisty Intersection Signature Trait (GIST), a novel approach that enhances Graph Transformers by explicitly encoding graph structures. GIST captures substructures through pairwise node intersection estimates and incorporates this information as an attention bias, enabling more effective modeling of structural relationships. Our theoretical analysis and empirical evaluations demonstrate that GIST preserves key structural information essential for graph-level task. Across diverse benchmark datasets, Graph Transformers augmented with GIST maintains a consistently strong performance profile. These results underscore the value of structure-aware attention in advancing graph representation learning and fostering more robust and interpretable models for scientific applications.

## ETHICS STATEMENT

Given the technical focus of this work on algorithmic improvements for structural encoding in Graph Transformers, we do not identify specific limitations that require emphasis within the scope of our methodology. The design and evaluation of GIST are grounded in theoretical analysis and controlled benchmarking, and the method demonstrates robust performance across diverse graph-level tasks. Regarding societal impact, this work does not introduce novel data, application-specific deployments, or user-facing components. As such, there are no direct negative societal consequences inherent to the algorithm itself. Any potential downstream effects would depend on the specific applications in which GIST is integrated—for example, in domains like drug discovery or social network analysis—where ethical considerations may vary by context. We encourage responsible usage aligned with domain-specific best practices.

## REPRODUCIBILITY STATEMENT

We facilitate reproducibility through multiple artifacts and detailed documentation. A repository will be released in the future host source code, experiment scripts, and configuration files. . Experiment details and method details for implementation are included in Section 3, Section 4, Section 5, and Appendix E.

Public release of the code is temporarily deferred to comply with institutional policies governing the dissemination of software and research artifacts (including reviews for IP ownership, confidentiality, and third-party licensing). We are securing the necessary authorizations and access credentials for an open-source release. Upon the paper's publication, we will promptly make the repository available under an appropriate license.

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

## APPENDIX

## A    USAGE OF LARGE LANGUAGE MODELS

Large language models were used exclusively for minor text refinements, such as paraphrasing and improving fluency. All outputs were reviewed and corrected by the authors, and the scientific contributions remain fully original and author-driven.

## B    RELATED WORKS

**Graph Substructures Modeling.** Modeling graph substructures is crucial for capturing fine-grained structural patterns and improving representation learning in graph-based tasks. However, GNNs remain fundamentally constrained by their reliance on localized message passing, which limits their ability to capture long-range dependencies and effectively model complex substructure interactions, due to over-smoothing and over-squashing issues (Xu et al., 2018; Alon & Yahav, 2021). To address this, later works have introduced spectral features (Balcilar et al., 2021), motif-based methods (Rong et al., 2020; Zhang et al., 2021; Bar-Shalom et al., 2024; Wollschlager et al., 2024), and Weisfeiler-Lehman (WL) kernel-based approaches (Morris et al., 2019) to improve graph representation learning by explicitly capturing local and global structural patterns. While motif-based methods improve expressivity by incorporating recurring substructures, they often depend on predefined motifs, restricting their adaptability to unseen graph patterns. Similarly, WL kernel-based approaches enhance structural discrimination but struggle with distinguishing graphs that are structurally different yet WL-equivalent. Furthermore, spectral features capture global graph properties but introduce additional computational complexity, making them less practical for large-scale applications. These limitations underscore the need for alternative architectures that can more effectively integrate structural biases while maintaining both scalability and expressiveness in graph learning.

**Graph Transformers.** Transformers have demonstrated remarkable success in natural language processing and computer vision by leveraging self-attention to model long-range dependencies effectively (Vaswani et al., 2017). More recently, their adaptation to graph-structured data has led to the emergence of Graph Transformers, where self-attention replaces traditional message-passing mechanisms to enable more flexible and expressive learning (Zhang et al., 2020; Dwivedi & Bresson, 2021). However, a fundamental challenge in applying Transformers to graphs is the absence of a natural node ordering, making it difficult to encode structural information directly. To address this, positional encodings have been introduced to assign meaningful node representations within the graph topology. Among these, Laplacian eigenvector-based encodings (LapPE) (Dwivedi et al., 2022a) and random walk positional encodings (RWPE) (Dwivedi et al., 2022b) inject global structural awareness, enhancing the model's ability to differentiate nodes with similar local neighborhoods. Beyond positional encodings, researchers have explored incorporating structural biases into self-attention to ensure that Graph Transformers respect the underlying graph topology. GPS (Rampášek et al., 2022) combines message passing with attention, allowing models to capture both local and global dependencies within the graph. More recently, GRIT (Ma et al., 2023) introduced a fully Transformer-based framework that eliminates explicit message passing and embeds structure-aware attention with RRWP, while Airale et al. (2025) introduces a new structural encoding method with estimation on the number of simple paths between nodes. These advancements reflect a growing shift toward pure Transformer architectures that effectively incorporate graph-specific inductive biases, paving the way for more scalable and expressive models in graph representation learning.

# C PROOFS

## C.1 GIST EXPRESSIVENESS

We first recall some relevant definitions. Let $\mathcal{G} = (\mathcal{V}, \mathcal{E})$ be an undirected graph. We use $d^{\text{SP}}(u, v)$ to denote the shortest path distance from node $u$ to node $v$. For every node $u \in \mathcal{V}$, we write $\mathcal{N}(u)$ for its direct neighbors in $\mathcal{G}$, and denote its $k$-hop neighborhoods as $\mathcal{N}_k(u)$: it consists of all nodes whose shortest path distances from $u$ are less than or equal to $k$. Additionally, we define the $k$-hop common neighborhood of a node pair $(u, v)$ as $\mathcal{C}_{k_u, k_v}(u, v) = \mathcal{N}_{k_u}(u) \cap \mathcal{N}_{k_v}(v)$, which is the set of nodes in the graph that are within $k_u$-hop from $u$ and with $k_v$-hop from $v$, respectively. The *diameter* of $\mathcal{G}$, $D(\mathcal{G}) = \max_{u,v} d^{\text{SP}}(u, v)$, is the maximum shortest path distance between any pair of nodes.

$\mathcal{G}$ is called *distance-regular* if for all $1 \le i, j \le D(\mathcal{G})$ and for all nodes $u, v, x, y \in \mathcal{V}$ with $d^{\text{SP}}(u, v) = d^{\text{SP}}(x, y)$, we have $|\mathcal{C}_{i,j}(u, v)| = |\mathcal{C}_{i,j}(x, y)|$. In other words, for any two nodes $u$ and $v$, the number of nodes at distance $i$ from $u$ and at distance $j$ from $v$ depends only on $i, j$, and the distance between $u$ and $v$. It follows immediately that, for all $u, v \in \mathcal{V}$ and $1 \le i \le D(\mathcal{G})$, $|\mathcal{N}_i(u)| = |\mathcal{N}_i(v)|$, i.e., the number of $i$-hop neighbors is the same for all nodes. We thus can define $\kappa(\mathcal{G}) = (k_1, \ldots, k_{D(\mathcal{G})})$ as the $k$-hop-neighbor array where $k_i := |\mathcal{N}_i(u)|$ for *every* $u \in \mathcal{V}$. Furthermore, the *intersection array* of a distance-regular graph $\mathcal{G}$ is defined by $\iota(\mathcal{G}) = (b_0, \ldots, b_{D(\mathcal{G})-1}; c_1, \ldots, c_{D(\mathcal{G})})$ which, for every $1 \le j \le D(\mathcal{G})$ and every pair of nodes $u, v \in \mathcal{V}$ with $d^{\text{SP}}(u, v) = j$, specifies that $|\mathcal{N}(u) \cap \mathcal{N}_{j+1}(v)| = b_j$ and $|\mathcal{N}(u) \cap \mathcal{N}_{j-1}(v)| = c_j$.

The *effective resistance distance* between a pair of node $u, v \in \mathcal{V}$ is defined as follows. Identify $\mathcal{G} = (\mathcal{V}, \mathcal{E})$ with an electrical network on $n$ nodes in which each edge corresponds to a link of unit conductance. If we inject a unit of current into $u$ and extract a unit of current from $v$, then the induced voltage difference between nodes $u$ and $v$ is defined as the effective resistance between these two nodes, denoted $d^{\text{RD}}(u, v)$. One can show that effective resistance indeed defines distance metric on $\mathcal{V} \times \mathcal{V}$: $d^{\text{RD}}(\cdot, \cdot)$ is non-negative, semidefinite, symmetric, and satisfies the triangle inequality. It is well-known that the $n \times n$ *resistance distance matrix*, whose $(u, v)$-entry is $d^{\text{RD}}(u, v)$, can be computed by the Moore-Penrose inverse of the Laplacian of $\mathcal{G}$, see e.g. Theorem E.1 in Zhang et al. (2023b).

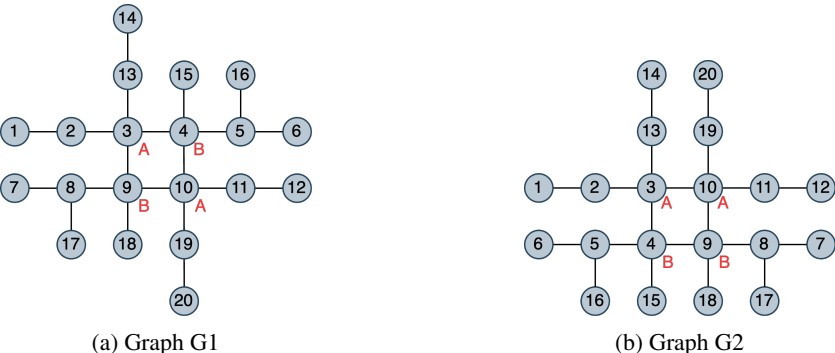

(a) Graph G1        (b) Graph G2

Figure 3: Graph pair that GIST can distinguish while RD-WL and (truncated) RRWP can't

**Theorem C.1** (Restatement of Theorem 3.3). *For the expressive power of GIST, we have the following:*

1. *GIST($n - 1$) is more expressive than SPD-WL.*

2. *There exists a pair of graphs such that GIST($n - 1$) distinguishes them while RD-WL does not.*

3. *There exist a pair of graphs such that GIST distinguishes them while RRWP does not.*

*Proof.* **Item 1.** To see that GIST($n - 1$) is as expressive as SPD-WL, observe that when $k = n - 1$, $\mathcal{I}_{k_u, k_v}(u, v)$ encodes the counts of all nodes in $\mathcal{V}$ indexed by their distance from $u$ and $v$. In particular, $d^{\text{SP}}(u, v)$ can be read out readily as $d^{\text{SP}}(u, v) = 1 + \min_i\{\mathcal{I}_{1,i}(u, v) > 0\}$, as $\mathcal{I}_{1,i}(u, v)$ counts the

number of nodes at distance 1 from $u$ and at distance $i$ from $v$, and if $d^{\text{SP}}(u, v) = i+1$, then any node on the shortest path between $u$ and $v$ whose distance is 1 from $u$ satisfies this condition. Therefore, by aggregating the counts of this method over all vertices $v \in \mathcal{V}$, we can easily get from GIST the shortest path counts encoded in SPD-WL. To see that GIST$(n-1)$ is more expressive than SPD-WL, we employ a theorem proved in Zhang et al. (2023b) (Theorem C.58), which states that SPD-WL can distinguish two distance-regular graphs $G$ and $H$ if and only if their $k$-hop-neighbor arrays differ, i.e. $\kappa(G) \neq \kappa(H)$. Note that for distance regualr graphs, GIST encodes both the $k$-hop-neighbor arrays and the intersection array. Consequently, as demonstrated in Zhang et al. (2023b), SPD-WL fails to distinguish between the Dodecahedron graph and the Desargues graph while GIST can.

**Item 2.** We conjecture that there are some graphs that RD-WL can distinguish while GIST$(n-1)$ can't, i.e. these two encoding schemes are incomparable. We present a graph pair for which GIST$(n-1)$ is more expressive than RD-WL. Such a graph pair is shown in Fig. 3. One can verify that 2-WL (or equivalently 1-FWL) would color the 20 nodes of both graphs into seven color classes. As demonstrated in Table 9, augmenting with resistance distance fails to distinguish between graphs G1 and G2. On the other hand, as shown in Table 7 and Table 8, the GIST node signatures of node class [3, 10] are distinct for graphs G1 and G2.

**Item 3.** We use the graph pair in Fig. 3 again. As shown in Table 10, if we use a truncated RRWP (specifically by setting $k = 3$), RRWP(3) can not distinguish between G1 and G2. On the other hand, as shown in Item 2, GIST can successfully distinguish between these two graphs. As the diameter of both graphs is 6, our example thus shows that GIST(6) is more expressive than RRWP(3) for certain class of graphs. $\qquad\square$

Table 7: GIST-Signature-to-Node Mapping on Graph G1

| GIST Signature $(S_k(u, v)$, count) | Node IDs |
|---|---|
| {((0,0,1,0,1,2,1,2,5),5), ((0,1,1,1,2,2,1,4,4),3), ((1,1,1,1,1,1,4,4,4),1), ((0,0,0,0,0,1,0,2,3),2), ((0,0,0,0,0,1,0,1,2),4), ((0,0,0,1,1,1,1,4,4),1), ((0,0,1,0,1,2,2,3,5),1), ((0,0,0,0,0,0,0,0,2),2)} | [1, 12, 14, 20] |
| {((0,1,1,3,4,4,3,10,10),1), ((0,1,1,1,2,4,1,4,6),4), ((0,0,1,0,1,2,1,2,5),4), ((0,0,0,0,0,2,0,1,3),2), ((1,1,2,1,3,4,4,6,10),2), ((0,1,1,0,1,4,0,1,4),1), ((0,1,1,2,3,4,2,7,8),1), ((1,1,2,1,3,4,2,4,10),1), ((0,1,1,1,2,4,1,2,4),1), ((0,0,1,0,2,3,1,3,8),2)} | [2, 11, 13, 19] |
| {((0,2,2,1,3,7,1,4,8),2), ((1,1,4,1,3,6,3,5,10),2), ((1,1,4,1,1,4,1,1,4),2), ((0,0,2,0,1,3,1,2,5),2), ((2,2,4,2,6,8,4,8,14),1), ((1,1,4,1,3,6,1,3,10),2), ((0,3,3,3,6,10,3,10,14),2), ((0,1,1,1,2,4,1,3,5),4), ((0,3,3,1,4,10,1,4,10),2)} | [3, 10] |
| {((0,2,2,1,3,7,1,4,8),2), ((1,1,4,1,3,6,2,4,10),4), ((0,0,2,0,1,3,1,2,5),2), ((0,3,3,0,3,10,0,3,10),1), ((0,3,3,2,5,10,2,5,10),1), ((2,2,4,2,6,8,4,8,14),1), ((1,1,4,1,2,5,1,2,5),2), ((0,3,3,3,6,10,3,10,14),2), ((0,1,1,1,2,4,1,2,4),4)} | [4, 9] |
| {((0,1,1,1,2,4,1,4,6),4), ((0,0,1,0,1,2,1,2,5),4), ((0,2,2,0,2,5,0,2,5),2), ((0,0,0,0,0,2,0,1,3),2), ((1,1,3,1,3,5,1,3,10),1), ((0,2,2,3,5,5,3,10,10),1), ((0,1,1,2,3,4,2,7,8),1), ((1,1,3,1,3,5,4,6,10),2), ((0,0,1,0,2,3,1,3,8),2)} | [5, 8] |
| {((0,1,1,1,2,3,1,4,5),3), ((1,1,1,1,2,2,1,2,5),1), ((0,0,1,0,1,2,1,2,5),4), ((1,1,1,1,2,2,4,5,5),1), ((0,0,0,0,0,1,0,2,3),2), ((0,0,0,0,0,1,0,1,2),4), ((0,0,0,2,2,2,2,5,5),1), ((0,0,1,0,1,2,2,3,5),1), ((0,0,0,0,0,0,0,0,2),2)} | [6, 7, 16, 17] |
| {((0,0,0,3,3,3,3,10,10),1), ((0,0,1,1,2,4,1,4,6),4), ((1,1,1,1,3,3,3,5,10),1), ((1,1,1,1,3,3,4,6,10),2), ((0,0,1,0,1,2,1,2,5),4), ((0,0,0,0,0,2,0,1,3),2), ((0,1,1,1,2,4,1,3,5),2), ((0,1,1,2,3,4,2,7,8),1), ((0,0,1,0,2,3,1,3,8),2)} | [15, 18] |

Table 8: GIST-Signature-to-Node Mapping on Graph G2

| GIST Signature $(S_k(u, v)$, count) | Node IDs |
|---|---|
| {((0,0,1,0,1,2,1,2,5),5), ((0,1,1,1,2,2,1,4,4),3), ((1,1,1,1,1,1,4,4,4),1), ((0,0,0,0,0,1,0,2,3),2), ((0,0,0,0,0,1,0,1,2),4), ((0,0,0,1,1,1,1,4,4),1), ((0,0,1,0,1,2,2,3,5),1), ((0,0,0,0,0,0,0,0,2),2)} | [1, 12, 14, 20] |
| {((0,1,1,3,4,4,3,10,10),1), ((0,1,1,1,2,4,1,4,6),4), ((0,0,1,0,1,2,1,2,5),4), ((0,0,0,0,0,2,0,1,3),2), ((1,1,2,1,3,4,4,6,10),2), ((0,1,1,0,1,4,0,1,4),1), ((0,1,1,2,3,4,2,7,8),1), ((1,1,2,1,3,4,2,4,10),1), ((0,1,1,1,2,4,1,2,4),1), ((0,0,1,0,2,3,1,3,8),2)} | [2, 11, 13, 19] |
| {((0,2,2,1,3,7,1,4,8),2), ((1,1,4,1,1,4,1,1,4),2), ((0,0,2,0,1,3,1,2,5),2), ((0,1,1,1,2,4,1,2,4),2), ((0,1,1,1,2,4,1,3,5),2), ((2,2,4,2,6,8,4,8,14),1), ((1,1,4,1,3,6,3,5,10),1), ((0,3,3,3,6,10,3,10,14),2), ((1,1,4,1,3,6,2,4,10),2), ((1,1,4,1,3,6,1,3,10),1), ((0,3,3,1,4,10,1,4,10),2)} | [3, 10] |
| {((0,2,2,1,3,7,1,4,8),2), ((0,0,2,0,1,3,1,2,5),2), ((0,3,3,0,3,10,0,3,10),1), ((0,3,3,2,5,10,2,5,10),1), ((0,1,1,1,2,4,1,2,4),2), ((0,1,1,1,2,4,1,3,5),2), ((2,2,4,2,6,8,4,8,14),1), ((1,1,4,1,3,6,3,5,10),1), ((1,1,4,1,2,5,1,2,5),2), ((1,1,4,1,3,6,2,4,10),2), ((0,3,3,3,6,10,3,10,14),2), ((1,1,4,1,3,6,1,3,10),1)} | [4, 9] |
| {((0,1,1,1,2,4,1,4,6),4), ((0,0,1,0,1,2,1,2,5),4), ((0,2,2,0,2,5,0,2,5),2), ((0,0,0,0,0,2,0,1,3),2), ((1,1,3,1,3,5,1,3,10),1), ((0,2,2,3,5,5,3,10,10),1), ((0,1,1,2,3,4,2,7,8),1), ((1,1,3,1,3,5,4,6,10),2), ((0,0,1,0,2,3,1,3,8),2)} | [5, 8] |
| {((0,1,1,1,2,3,1,4,5),3), ((1,1,1,1,2,2,1,2,5),1), ((0,0,1,0,1,2,1,2,5),4), ((1,1,1,1,2,2,4,5,5),1), ((0,0,0,0,0,1,0,2,3),2), ((0,0,0,0,0,1,0,1,2),4), ((0,0,0,2,2,2,2,5,5),1), ((0,0,1,0,1,2,2,3,5),1), ((0,0,0,0,0,0,0,0,2),2)} | [6, 7, 16, 17] |
| {((0,0,0,3,3,3,3,10,10),1), ((0,0,1,1,2,4,1,4,6),4), ((1,1,1,1,3,3,3,5,10),1), ((1,1,1,1,3,3,4,6,10),2), ((0,0,1,0,1,2,1,2,5),4), ((0,0,0,0,0,2,0,1,3),2), ((0,1,1,1,2,4,1,3,5),2), ((0,1,1,2,3,4,2,7,8),1), ((0,0,1,0,2,3,1,3,8),2)} | [15, 18] |

Table 9: Resistance Distance Signature-to-Node Mapping on Graph G1 and G2

| Resistance Distance Signature (value, count) | Node IDs |
|---|---|
| {((4.0,), 3), ((3.75,), 4), ((2.75,), 2), ((2.0,), 1), ((1.0,), 1), ((5.0,), 2), ((3.0,), 2), ((4.75,), 4)} | [1, 12, 14, 20] |
| {((1.0,), 2), ((2.0,), 2), ((3.0,), 3), ((2.75,), 4), ((1.75,), 2), ((3.75,), 4), ((4.0,), 2)} | [2, 11, 13, 19] |
| {((1.75,), 4), ((1.0,), 3), ((2.75,), 4), ((0.75,), 2), ((2.0,), 4), ((3.0,), 2)} | [3, 4, 9, 10] |
| {((2.0,), 2), ((2.75,), 4), ((1.0,), 3), ((3.0,), 2), ((1.75,), 2), ((3.75,), 4), ((4.0,), 2)} | [5, 8] |
| {((2.0,), 2), ((4.75,), 4), ((2.75,), 2), ((1.0,), 1), ((5.0,), 2), ((4.0,), 2), ((3.0,), 2), ((3.75,), 4)} | [6, 7, 16, 17] |
| {((2.0,), 2), ((2.75,), 4), ((1.75,), 2), ((1.0,), 1), ((3.0,), 4), ((3.75,), 4), ((4.0,), 2)} | [15, 18] |

Table 10: RRWP with (truncated) $k$ Signature-to-Node Mapping on Graph G1 and G2

| RRWP Signature (vector, count) | Node IDs |
|---|---|
| {((0.0, 0.0, 0.5), 1), ((0.0, 1.0, 0.0), 1), ((0.0, 0.0, 0.0), 17)} | [1, 12, 14, 20] |
| {((0.0, 0.0, 0.125), 3), ((0.0, 0.0, 0.0), 14), ((0.0, 0.5, 0.0), 2)} | [2, 11, 13, 19] |
| {((0.0, 0.0, 0.125), 3), ((0.0, 0.0, 0.0), 8), ((0.0, 0.0, 0.0625), 4), ((0.0, 0.25, 0.0), 4)} | [3, 10] |
| {((0.0, 0.0, 0.0625), 4), ((0.0, 0.0, 0.0), 8), ((0.0, 0.0, 0.083333), 2), ((0.0, 0.0, 0.125), 1), ((0.0, 0.25, 0.0), 4)} | [4, 9] |
| {((0.0, 0.0, 0.0), 13), ((0.0, 0.0, 0.083333), 3), ((0.0, 0.333333, 0.0), 3)} | [5, 8] |
| {((0.0, 0.0, 0.333333), 2), ((0.0, 1.0, 0.0), 1), ((0.0, 0.0, 0.0), 16)} | [6, 7, 16, 17] |
| {((0.0, 1.0, 0.0), 1), ((0.0, 0.0, 0.25), 3), ((0.0, 0.0, 0.0), 15)} | [15, 18] |

## C.2 GIST INVARIANCE

**Proposition C.2.** *Let $\mathcal{G} = (\mathcal{V}, \mathcal{E})$ denote a graph with $n$ nodes ($|\mathcal{V}| = n$). Let $S_k(u, v) \in \mathbb{R}^{k^2 + 2k}$ denote the $k$-hop GIST encoding of every ordered node pair $(u, v)$ (see Definition 3.2). Then the GIST attention as defined in Definition 4.2, $\{\{\psi(x_u) : u \in \mathcal{V}\}\}$, is invariant under graph isomorphism.*

*Proof.* This follows directly from the fact that both the initial node representation $\{x_u : u \in \mathcal{V}\}$ and the initial edge representation $\{y_{u,v} : (u, v) \in \mathcal{E}\}$ are isomorphism invariant, together with the fact that, since its encoding $S_k(u, v)$ only *counts* the number of nodes of various distances from $u$ and $v$, GIST representation $\{S_k(u, v) : (u, v) \in \mathcal{V} \times \mathcal{V}\}$ is also isomorphism invariant. It follows that if $f$ is any isomorphism between graph $\mathcal{G}$ and $\mathcal{H}$, we have that for every $u \in V(\mathcal{G})$, $x_{f(u)} = x_u$ and for any node pair $(u, v) \in V(\mathcal{G}) \times V(\mathcal{G})$, $y_{f(u),f(v)} = y_{u,v}$ and $S_k(f(u), f(v)) = S_k(u, v)$, hence $\psi(x_{f(u)}) = \psi(x_u)$ for every $u \in V(\mathcal{G})$. $\qquad\square$

## C.3 ESTIMATION VARIANCE OF GIST WITH HASHING

We approximate the $k$-hop common neighborhood size $|\mathcal{C}_{k_u, k_v}(u, v)|$ using MinHash and Hyper-LogLog sketches. For a fixed node pair $(u, v)$ and hops $(k_u, k_v)$, recall that

$$|\mathcal{C}_{k_u, k_v}(u, v)| = J_{k_u, k_v}(u, v) \cdot U_{k_u, k_v}(u, v),$$

where $J_{k_u, k_v}(u, v)$ is the Jaccard similarity between $\mathcal{N}_{k_u}(u)$ and $\mathcal{N}_{k_v}(v)$, and $U_{k_u, k_v}(u, v) = |\mathcal{N}_{k_u}(u) \cup \mathcal{N}_{k_v}(v)|$. For brevity we write

$$J := J_{k_u, k_v}(u, v), \qquad U := U_{k_u, k_v}(u, v), \qquad C := |\mathcal{C}_{k_u, k_v}(u, v)| = J U.$$

We estimate $J$ with MinHash using $m$ hash functions, and $U$ with HyperLogLog using precision parameter $p$ and $m_{\mathrm{HLL}} = 2^p$ registers. Let $\widehat{J}$ and $\widehat{U}$ denote the corresponding estimators and define

$$\widehat{C} := \widehat{J}\widehat{U}$$

as the estimator of $C$.

**Unbiasedness.**  Standard properties of MinHash and HyperLogLog yield

$$\mathbb{E}[\widehat{J}] = J, \qquad \operatorname{Var}[\widehat{J}] = \frac{J(1-J)}{m},$$

$$\mathbb{E}[\widehat{U}] = U, \qquad \operatorname{Var}[\widehat{U}] = U^2 \alpha_p^2,$$

where $\alpha_p = \Theta(1/\sqrt{m_{\mathrm{HLL}}})$ is the usual HyperLogLog constant.

Since MinHash and HyperLogLog use independent hash functions, $\widehat{J}$ and $\widehat{U}$ are independent, so

$$\mathbb{E}[\widehat{C}] = \mathbb{E}[\widehat{J}\widehat{U}] = \mathbb{E}[\widehat{J}]\,\mathbb{E}[\widehat{U}] = J\,U = C.$$

Thus $\widehat{C}$ is an unbiased estimator of $|\mathcal{C}_{k_u,k_v}(u,v)|$.

**Variance bound.**  We first record an elementary identity for the variance of a product of independent random variables.

**Lemma C.3** (Variance of a product). *Let $X$ and $Y$ be independent random variables with finite second moments. Then*

$$\operatorname{Var}(XY) = \mathbb{E}[X^2]\operatorname{Var}(Y) + \mathbb{E}[Y]^2\operatorname{Var}(X).$$

*Proof.*  By definition,

$$\operatorname{Var}(XY) = \mathbb{E}[X^2 Y^2] - \big(\mathbb{E}[XY]\big)^2.$$

Independence implies $\mathbb{E}[X^2 Y^2] = \mathbb{E}[X^2]\mathbb{E}[Y^2]$ and $\mathbb{E}[XY] = \mathbb{E}[X]\mathbb{E}[Y]$, so

$$\operatorname{Var}(XY) = \mathbb{E}[X^2]\mathbb{E}[Y^2] - \mathbb{E}[X]^2\mathbb{E}[Y]^2.$$

We add and subtract $\mathbb{E}[X^2]\mathbb{E}[Y]^2$:

$$\operatorname{Var}(XY) = \mathbb{E}[X^2]\big(\mathbb{E}[Y^2] - \mathbb{E}[Y]^2\big) + \mathbb{E}[Y]^2\big(\mathbb{E}[X^2] - \mathbb{E}[X]^2\big)$$
$$= \mathbb{E}[X^2]\operatorname{Var}(Y) + \mathbb{E}[Y]^2\operatorname{Var}(X),$$

as claimed. $\square$

We apply Lemma C.3 with $X = \widehat{J}$ and $Y = \widehat{U}$:

$$\operatorname{Var}(\widehat{C}) = \operatorname{Var}(\widehat{J}\widehat{U})$$
$$= \mathbb{E}[\widehat{J}^2]\operatorname{Var}(\widehat{U}) + \mathbb{E}[\widehat{U}]^2\operatorname{Var}(\widehat{J}).$$

Using the variance identity $\mathbb{E}[\widehat{J}^2] = \operatorname{Var}[\widehat{J}] + (\mathbb{E}[\widehat{J}])^2$ and $\mathbb{E}[\widehat{U}] = U$, we obtain

$$\operatorname{Var}(\widehat{C}) = \big(\operatorname{Var}[\widehat{J}] + (\mathbb{E}[\widehat{J}])^2\big)\operatorname{Var}[\widehat{U}] + U^2\operatorname{Var}[\widehat{J}]$$
$$= \left(\frac{J(1-J)}{m} + J^2\right)U^2\alpha_p^2 + U^2\frac{J(1-J)}{m}$$
$$= U^2\left[J^2\alpha_p^2 + \frac{J(1-J)}{m}(1+\alpha_p^2)\right].$$

Since $0 \le J \le 1$ and $J(1-J) \le 1/4$, we have the simple upper bound

$$\operatorname{Var}(\widehat{C}) \;\le\; U^2\left[\alpha_p^2 + \frac{1+\alpha_p^2}{4m}\right].$$

Because $\alpha_p = \Theta(1/\sqrt{m_{\mathrm{HLL}}})$, this shows

$$\operatorname{Var}(\widehat{C}) = O\left(U^2\left[\frac{1}{m_{\mathrm{HLL}}} + \frac{1}{m}\right]\right),$$

so increasing either the number of MinHash functions $m$ or the HLL precision $p$ reduces the variance.

**Corollary C.4** (Relative error of the GIST estimator). *Assume $J > 0$ (i.e., the $k$-hop common neighborhood is non-empty). Then the squared coefficient of variation of $\widehat{C}$ is*

$$\frac{\mathrm{Var}(\widehat{C})}{C^2} = \frac{\mathrm{Var}(\widehat{C})}{J^2 U^2} = \alpha_p^2 + \frac{(1 - J)(1 + \alpha_p^2)}{m\,J} \;\leq\; \alpha_p^2 + \frac{1 + \alpha_p^2}{m\,J},$$

*and hence*

$$\mathrm{CV}(\widehat{C}) := \frac{\sqrt{\mathrm{Var}(\widehat{C})}}{C} \;\leq\; \alpha_p + \sqrt{\frac{1 + \alpha_p^2}{m\,J}}.$$

*In particular, for any fixed $J > 0$ we have*

$$\mathrm{CV}(\widehat{C}) = O\!\left(\frac{1}{\sqrt{m_{\mathrm{HLL}}}} + \frac{1}{\sqrt{m}}\right),$$

*so the relative error of the GIST estimator decreases at the standard Monte Carlo rate in both the number of MinHash functions and the number of HyperLogLog registers.*

This formalizes that the GIST estimator $\widehat{C}_{k_u,k_v}(u,v)$ is unbiased and admits a controllable relative error.

## D   GIST ESTIMATION ALGORITHM

We present the GIST estimation algorithm in Algorithm 1 and Algorithm 2.

---

**Algorithm 1** Algorithm for computing intersection cardinality $|\mathcal{C}_{k_u,k_v}(u,v)|$

---

**Input:** Graph $\mathcal{G} = (\mathcal{V}, \mathcal{E})$, max hops $k$, hops $k_u, k_v$, $m$ MinHash functions $H = \{h_1, \ldots, h_m\}$, HyperLogLog parameter $p$ and regularizer constant $\alpha_p$
**Output:** Intersection cardinality $|\mathcal{C}_{k_u,k_v}(u,v)|$
{Step 1. Pre-compute MinHash signatures}
**for** $v \in \mathcal{V}, h_j \in H$ **do**
    $M_v[j,0] \leftarrow h_j(v)$ {Initialize MinHash signatures}
**end for**
**for** $i = 1$ **to** $k$ **do**
    **for** $v \in \mathcal{V}, h_j \in H$ **do**
        $M_v[j,i] \leftarrow \min\limits_{u \in \mathcal{N}(v)} \big(M_u[j,i-1], M_v[j,i-1]\big)$
    **end for**
**end for**
{Step 2. Pre-compute HyperLogLog sketches}
$m \leftarrow 2^p$
**for** $v \in \mathcal{V}$ **do**
    Compute $k$-hop HyperLogLog sketch $H_v \in \mathbb{R}^{m \times k}$
**end for**
{Step 3. Compute intersection cardinality}
**for** $(u,v) \in \mathcal{V} \times \mathcal{V}$ **do**
    $\tilde{\mathcal{J}}_{k_u,k_v}(u,v) \leftarrow$ JACCARD-EST$(k_u, k_v, m, M_u, M_v)$
    $\tilde{\mathcal{U}}_{k_u,k_v}(u,v) \leftarrow$ HLL-EST$(k_u, k_v, H_u, H_v)$
    $|\mathcal{C}_{k_u,k_v}(u,v)| \leftarrow \tilde{\mathcal{J}}_{k_u,k_v}(u,v) \cdot \tilde{\mathcal{U}}_{k_u,k_v}(u,v)$
**end for**
**return** $|\mathcal{C}_{k_u,k_v}(u,v)|$

**Function:** JACCARD-EST$(k_u, k_v, m, M_u, M_v)$
**Input:** hops $k_u, k_v$, number of MINHASH functions $m$, and $k-$hop MinHash values $M_u, M_v$
**Output:** Jaccard similarity $\tilde{\mathcal{J}}_{k_u,k_v}(u,v)$
$\tilde{\mathcal{J}}_{k_u,k_v}(u,v) \leftarrow 0$
**for** $j = 1$ **to** $m$ **do**
    **if** $M_u(j, k_u) = M_v(j, k_v)$ **then**
        $\tilde{\mathcal{J}}_{k_u,k_v}(u,v) \leftarrow \tilde{\mathcal{J}}_{k_u,k_v}(u,v) + 1$
    **end if**
**end for**
$\tilde{\mathcal{J}}_{k_u,k_v}(u,v) \leftarrow \tilde{\mathcal{J}}_{k_u,k_v}(u,v)/m$
**return** $\tilde{\mathcal{J}}_{k_u,k_v}(u,v)$
**EndFunction**

**Function:** HLL-EST$(k_u, k_v, H_u, H_v)$
**Input:** hops $k_u, k_v$, HyperLogLog sketches $H_u, H_v$
**Output:** Union cardinality $\tilde{\mathcal{U}}_{k_u,k_v}(u,v)$
$H_{k_u,k_v} \leftarrow \mathbf{0}^m$
**for** $j = 1$ **to** $m$ **do**
    $H_{k_u,k_v}[j] \leftarrow \max\big(H_u[j,k_u], H_v[j,k_v]\big)$
**end for**
$\tilde{\mathcal{U}}_{k_u,k_v}(u,v) \leftarrow \alpha_p m^2 (\sum_{i=0}^{m} 2^{-H_{k_u,k_v}[i]})^{-1}$
**return** $\tilde{\mathcal{U}}_{k_u,k_v}(u,v)$
**EndFunction**

---

---

**Algorithm 2** Computation of GIST structural encoding (using precomputed $|C_{k_u,k_v}(u,v)|$)

---

**Input:** Graph $\mathcal{G} = (\mathcal{V}, \mathcal{E})$, max hops $k$,
         intersection cardinalities $\widehat{C}_{k_u,k_v}(u,v)$ for all $(u,v) \in \mathcal{V} \times \mathcal{V}$ and $1 \le k_u, k_v \le k$,
         computed once by Alg. 1
**Output:** Pairwise GIST tensor $S_k \in \mathbb{R}^{|\mathcal{V}| \times |\mathcal{V}| \times (k^2+2k)}$ and/or node-wise summaries $\{\chi_u\}_{u \in \mathcal{V}}$

{Step 1. Get $k$-hop neighborhood sizes from diagonal $C_{t,t}(u,u)$}
**for** $u \in \mathcal{V}$ **do**
   **for** $t = 1$ **to** $k$ **do**
     $d_{u,t} \leftarrow \widehat{C}_{t,t}(u,u)$ {Since $C_{t,t}(u,u) = N_t(u)$ by definition}
   **end for**
**end for**

{Step 2. Compute internal and boundary counts}
**for** $(u,v) \in \mathcal{V} \times \mathcal{V}$ **do**
   Initialize $I_{k_u,k_v}(u,v) \leftarrow 0$ for all $1 \le k_u, k_v \le k$
   {(a) Internal counts via inclusion–exclusion (Def. 3.1)}
   **for** $k_u = 1$ **to** $k$ **do**
     **for** $k_v = 1$ **to** $k$ **do**
       $S \leftarrow 0$
       **for** $x = 1$ **to** $k_u$ **do**
         **for** $y = 1$ **to** $k_v$ **do**
           **if** $(x,y) \neq (k_u, k_v)$ **then**
             $S \leftarrow S + I_{x,y}(u,v)$
           **end if**
         **end for**
       **end for**
       $I_{k_u,k_v}(u,v) \leftarrow \widehat{C}_{k_u,k_v}(u,v) - S$
     **end for**
   **end for**
   {(b) Boundary counts from $u$- and $v$-sides}
   **for** $k_u = 1$ **to** $k$ **do**
     $B_{k_u,>k}(u,v) \leftarrow d_{u,k_u} - \sum_{k_v=1}^{k} I_{k_u,k_v}(u,v)$
   **end for**
   **for** $k_v = 1$ **to** $k$ **do**
     $B_{k_v,>k}(v,u) \leftarrow d_{v,k_v} - \sum_{k_u=1}^{k} I_{k_u,k_v}(u,v)$
   **end for**

   {Step 3. Form the GIST substructure vector $S_k(u,v)$}
   $S_k(u,v) \in \mathbb{R}^{k^2+2k} \leftarrow$ concatenate:
     (i) $\{I_{k_u,k_v}(u,v)\}_{1 \le k_u,k_v \le k}$ in a fixed order
     (ii) $\{B_{k_u,>k}(u,v)\}_{k_u=1}^{k}$
     (iii) $\{B_{k_v,>k}(v,u)\}_{k_v=1}^{k}$
**end for**

---

# E  Experiment Settings

**Dataset Statistics.** We provide the statistics of 12 datasets used in our experiments to evaluate the performance of our proposed GIST in Table 11.

Table 11: Datasets' Statistics

| Dataset | # Graphs | Avg. # nodes | Avg. # edges | Prediction task | Metric |
|---------|----------|--------------|--------------|-----------------|--------|
| BBBP | 2,050 | 23.9 | 51.6 | binary classification | ROC-AUC |
| Tox21 | 7,831 | 18.6 | 38.6 | 12-task classification | ROC-AUC |
| Toxcast | 8,597 | 18.7 | 38.4 | 617-task classification | ROC-AUC |
| Sider | 1,427 | 33.6 | 70.7 | 27-task classification | ROC-AUC |
| Clintox | 1,484 | 26.1 | 55.5 | 2-task classification | ROC-AUC |
| Bace | 1513 | 34.1 | 73.7 | binary classification | ROC-AUC |
| MUV | 93,087 | 24.2 | 52.6 | 17-task classification | ROC-AUC |
| HIV | 41,127 | 25.5 | 54.9 | binary classification | ROC-AUC |
| Peptides-func | 15,535 | 150.94 | 307.30 | 10-task classification | Avg. Precision |
| Peptides-struct | 15,535 | 150.94 | 307.30 | 11-task regression | Mean Abs. Error |
| Zinc Subset | 12,000 | 23.2 | 49.8 | regression | Mean Abs. Error |
| Zinc Full | 249,456 | 23.2 | 49.8 | regression | Mean Abs. Error |

**Baselines.** For each baseline, we either report the best results from existing literature or reproduce them using the official implementations with the hyperparameter settings specified in their respective papers. Specifically, for the MoleculeNet benchmark, we evaluate GRIT and GraphGPS across 8 datasets using their hyperparameters optimized for Peptides-struct, following Ma et al. (2023), which demonstrates that their performance is robust to different hyperparameter choices across datasets.

**LRGB Settings.** We follow the clear standard of benchmarking adopted in prior works like Ma et al. (2023); Bar-Shalom et al. (2024): for each dataset in the LRGB benchmark, we train our proposed method on the training split and select the model checkpoint that achieves the best validation performance. The corresponding test performance is then reported. Updated results of GCN(Kipf & Welling, 2017), GraphGPS(Rampášek et al., 2022), and GatedGCN(Dwivedi et al., 2022d) are taken directly from (Tonshoff et al., 2023).

**ZINC & ZINC-Full Settings.** We follow the common evaluation protocol established in prior works such as Dwivedi et al. (2022a); Ying et al. (2021): for both ZINC and ZINC-Full datasets, we train our model on the training split and select the checkpoint with the best validation performance. The test performance corresponding to this checkpoint is then reported.

**MoleculeNet Settings.** Following prior works such as (Luong & Singh, 2023; Liu et al., 2022), we adopt the scaffold-based splitting protocol provided by MoleculeNet for all datasets. Our model is trained on the training split, and the best checkpoint is selected based on validation performance. The corresponding test performance is then reported. Baseline results are either obtained directly from the original publications (e.g. (Luong & Singh, 2023; Liu et al., 2022)) or reproduced using their official code and best-reported hyperparameters (e.g. (Ma et al., 2023) or (Rampášek et al., 2022)).

# F  GIST ATTENTION EMPIRICAL STUDY

To better understand how GIST aids in distinguishing substructures within a graph and facilitates effective representation aggregation across them, we visualize the attention scores of Graph Transformers with and without GIST as the structural encoding. We further perform node clustering via spectral clustering using the learned GIST features on ZINC molecule graphs to examine whether structurally meaningful groupings emerge. The results indicate that, after integrating GIST, the attention mechanism tends to focus on coherent substructures—such as functional groups—rather than attending uniformly to individual nodes. This structured attention behavior highlights GIST's role in promoting both intra-substructure coherence and inter-substructure interaction, which are critical for accurate graph representation learning.

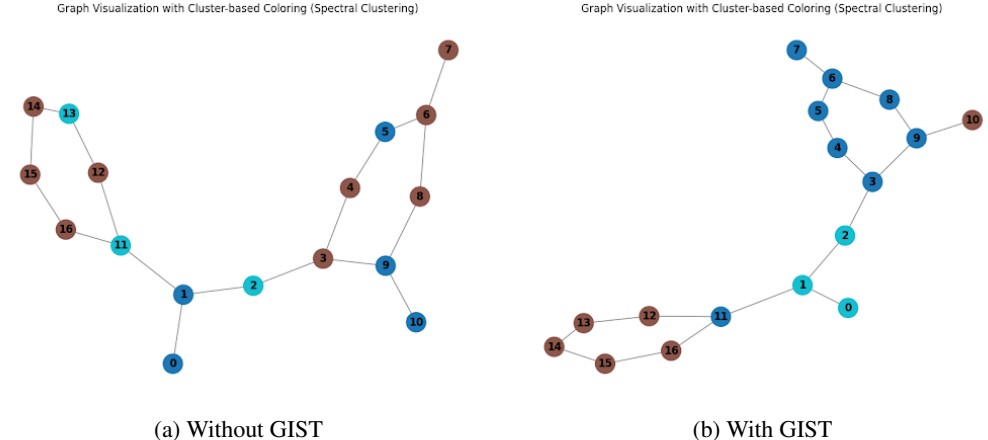

(a) Without GIST                    (b) With GIST

Figure 4: Clustering of Attention Scores on Graph 1
*To quantify how effectively GIST facilitates representation aggregation within and across substructures*, we analyze model attention on ZINC under a controlled backbone and evaluation protocol.

Using the same Graph Transformer backbone, we train three variants: (i) vanilla (no structural encoding), (ii) GRIT, and (iii) GIST. We then sample 1,000 graphs from the ZINC test set. For each graph, we partition nodes into substructures via the Louvain algorithm.

We define three complementary attention categories at the node–pair level after partitioning each graph into substructures. Let $C(u)$ denote the substructure (community) containing node $u$ and $s(u, v)$ denotes if there exists an edge between $u$ and $v$. For any ordered pair $(u, v)$, let $a(u, v)$ be the attention score from $u$ to $v$, We then bucketize attention score into three categories:

$$\textbf{Within-substructure:} \quad A_{\text{within}} = \frac{1}{|\{(u, v) : C(u) = C(v)\}|} \sum_{(u,v):\, C(u)=C(v)} a(u, v),$$

$$\textbf{Cross-substructure:} \quad A_{\text{cross}} = \frac{1}{|\{(u, v) : C(u) \neq C(v)\}|} \sum_{(u,v):\, C(u)\neq C(v)} a(u, v).$$

$$\textbf{Neighborhood:} \quad A_{\text{neighbor}} = \frac{1}{|\{(u, v) : s(u, v)\}|} \sum_{(u,v):\, s(u,v)} a(u, v).$$

Here, $A_{\text{within}}$ captures how strongly attention between pair of nodes from the same substructures, $A_{\text{cross}}$ captures attention allocated *between* different substructures and serves as a proxy for modeling higher-order interactions, whereas $A_{\text{neighbor}}$ captures how focusing the attention mechanism is on direct neighbors.

The community label $C(\cdot)$ is computed once per graph from topology alone and is held fixed across all methods. The designation "within-substructure" is defined per graph by the equality $C(u) = C(v)$; across different graphs, communities need not coincide as node sets to be regarded as comparable substructures—what matters is their structural form (e.g., isomorphic or structurally similar). Unless otherwise noted, self-pairs are excluded, and attention is aggregated by summing over heads and

Table 12: Attention Score of Within- vs. Cross-substructure and Neighbor among three different variants of Graph Transformer architecture

| **Variant** | $A_{\text{within}}$ | $A_{\text{cross}}$ | $A_{\text{neighbor}}$ |
|---|---|---|---|
| Vanilla | 0.50 | 0.44 | 0.71 |
| GRIT | 0.81 | 0.10 | 0.43 |
| GIST | 0.65 | 0.29 | 0.37 |

averaging across layers. For each model, we compute $A_{\text{within}}$, $A_{\text{cross}}$, and $A_{\text{neighbor}}$ on each of the 1,000 graphs and report their means in Table 12.

**Findings.** The vanilla Graph Transformer exhibits limited substructure awareness: its attention score is distributed nearly uniformly across node pairs while disproportionally attending to nearest neighbors, yielding negligible separation between within- and cross-substructure attention. GRIT displays the opposite pattern, concentrating attention predominantly on nodes with the same substructures (high within-substructure, low cross-substructure). By contrast, GIST maintains a *balanced* profile, allocating substantial attention both within substructures and across substructures—consistent with its design to capture substructures within a graph *and* their higher-order relationships.

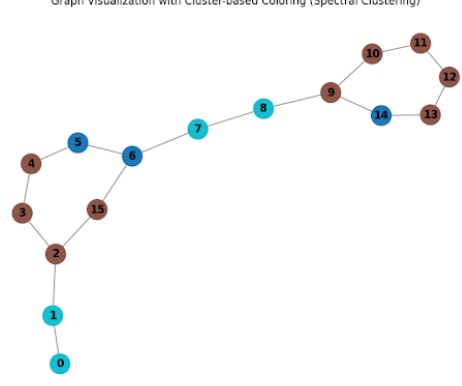

Figure 5: Clustering of Attention Scores with GIST

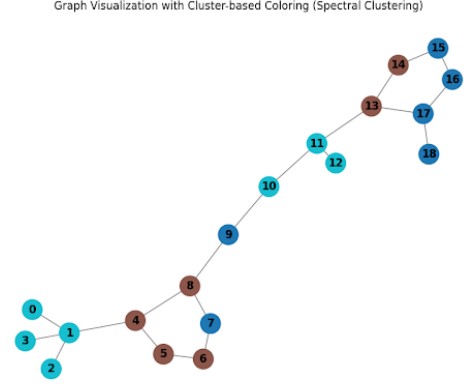

Figure 6: Clustering of Attention Scores with GIST

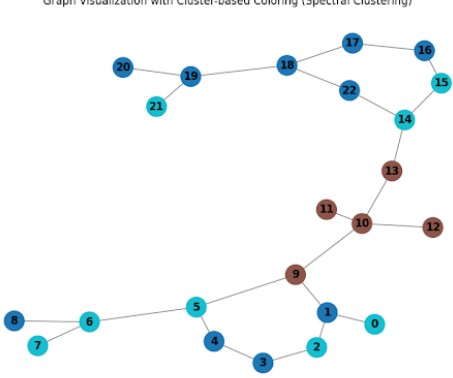

Figure 7: Clustering of Attention Scores with GIST

# G  ADDITIONAL EXPERIMENT RESULTS

## G.1  NODE-LEVEL AND LARGE-SCALE TASKS

We present the performance of GIST on Cluster, Pattern, and PCQM4Mv2 datasets in Table 13.

Table 13: Performance of GIST on Cluster, Pattern, and PCQM4Mv2

| Datasets | Cluster | Pattern | PCQM4Mv2 |
|---|---|---|---|
| GIST (ours) | 0.8196 | 0.8893 | 0.079 |
| GPS (Rampášek et al., 2022) | 0.7802 | 0.8668 | 0.094 |
| SAN (Kreuzer et al., 2021) | 0.7669 | 0.2486 | - |
| GRIT (Ma et al., 2023) | 0.8003 | 0.8720 | 0.086 |
| Exphormer (Shirzad et al., 2023) | 0.7807 | 0.8674 | - |
| SPSE (Airale et al., 2025) | 0.7957 | 0.8723 | 0.083 |
| CSA (Menegaux et al., 2024) | 0.7918 | 0.8701 | 0.085 |
| TIGT (Choi et al., 2024) | 0.7803 | 0.8668 | 0.083 |

## G.2  FULL-VERSION OF BASELINES' PERFORMANCE COMPARISON

Table 14: Performance on ZINC and Peptides Datasets. Reported as "Absolute Gap (Propotional Improvement %) between GIST's performance and the respective baseline's performance." + indicates GIST is better.

| Model | ZINC | ZINC-full | Peptides-struct | Peptides-func |
|---|---|---|---|---|
| GRIT | +0.009 (+15.2) | +0.004 (+17.4) | +0.0018 (+0.7) | -0.0005 (-0.07) |
| GPS | +0.020 (+28.6) | - | +0.0067 (+2.7) | +0.0449 (+6.9) |
| FragNet | +0.028 (+35.9) | +0.005 (+19.8) | +0.0020 (+0.8) | +0.0305 (+4.6) |
| Subgraphormer | +0.013 (+20.6) | +0.004 (+17.4) | +0.0052 (+2.1) | +0.0568 (+8.9) |
| TIGT | +0.007 (+12.3) | -0.005 (-26.3) | +0.0043 (+1.7) | +0.0304 (+4.6) |
| CSA | +0.006 (+10.7) | - | - | - |
| SPSE | +0.009 (+15.3) | - | +0.0007 (+0.3) | +0.0038 (+0.006) |
| GNN-SSWL+ | +0.020 (+28.6) | +0.003 (+13.6) | - | - |
| GCN | +0.317 (+86.4) | +0.094 (+83.2) | +0.0018 (+0.7) | +0.0123 (1.8) |
| Gated-GCN | +0.040 (+44.4) | +0.003 (+13.6) | +0.035 (+1.4) | +0.0218 (+3.2) |
| DS-GNN | +0.037 (+42.5) | - | - | - |
| Graphormer | +0.072 (+59.0) | +0.033 (+63.5) | - | - |

Table 15: Performance on Molecular Datasets. Reported as "Absolute Gap (Propotional Improvement %) between GIST's performance and the respective baseline's performance." + indicates GIST is better.

| Model | BBBP | Tox21 | Toxcast | Sider | Clintox | Bace | MUV | HIV |
|---|---|---|---|---|---|---|---|---|
| GRIT | +3.7 (+5.3) | +1.3 (+1.7) | +1.7 (+2.6) | +1.0 (+1.7) | +2.3 (+2.7) | +1.6 (+1.9) | -1.6 (-2.0) | -0.3 (0.0) |
| GPS | +17.4 (+31.0) | +5.8 (+8.1) | +6.7 (+11.1) | +1.2 (+1.8) | +9.0 (+11.4) | +14.5 (+20.3) | +10.3 (+15.8) | +11 (+15.8) |
| Subgraphormer | - | - | - | - | - | +1.6 (+3.4) | - | -3.4 (-4.1) |
| GraphFP | +1.6 (+2.0) | +3.2 (+4.3) | +3.4 (+5.3) | -2.3 (-3.6) | +3.5 (+4.1) | +5.5 (+6.8) | +0.1 (+0.1) | -1.0 (-1.0) |
| GraphMVP | +5.1 (+7.4) | +2.7 (+3.6) | +4.6 (+7.3) | -1.0 (-1.6) | +9.2 (+11.6) | +9.2 (+12.0) | +0.5 (+0.7) | +2.2 (+2.9) |
| DS-GNN | - | - | - | - | - | +7.59 (+9.7) | - | -2.13 (-2.7) |

# H  GIST EFFICIENCY

We present an efficiency experiment on training time in Table 16, which shows that GIST's training time is comparable to—or even lower than—that of other Graph Transformers. Notably, unlike many pre-trained graph models, GIST does not rely on extensive pretraining, yet still outperforms most of them on the MoleculeNet benchmarks.

We also analyze the one-time computation overhead of GIST features (Table 16). As emphasized in Section 3.4, this overhead is incurred only once at the beginning of training and remains minimal. Our method employs a lightweight randomized estimation procedure using MinHash and HyperLogLog to approximate $k$-hop substructure intersections with a constant number of operations. Both theoretical analysis and empirical evidence (Table 16) confirm that GIST achieves efficient computation without compromising structural expressiveness.

Table 16: One-time pre-computation and Training time of GIST (hour:min)

| Datasets | ZINC | ZINC-full | Peptides-struct | Peptides-func |
|---|---|---|---|---|
| GIST precomputation | 00:03 | 01:08 | 00:12 | 00:12 |
| GIST Training Time | 11:09 | 55:21 | 05:40 | 05:30 |
| GRIT Training + Precomputation Time | 16:30 | 104:57 | 07:15 | 06:42 |
| GraphGPS Training + Precomputation Time | 13:30 | - | - | - |
| SAN Training + Precomputation Time | 32:15 | - | - | - |

