# OpenReview forum: "Get the GIST of Graphs with Intersection Signature"
_ICLR.cc/2026/Conference — Submitted to ICLR 2026_

### Official Review · Reviewer_1jQu · 2025-10-30

**Soundness:** 3
**Presentation:** 1
**Contribution:** 2
**Rating:** 2
**Confidence:** 4

**Summary:**

The paper proposes GIST, a graph transformer architecture that encodes structure through pairwise intersections of k-hop neighborhoods. The motivation is to provide an alternative to distance- or Laplacian-based positional encodings by measuring structural overlap directly. Theoretical claims include improved expressiveness over SPD-WL, RD-WL, and RRWP, and invariance under graph isomorphism. Experiments show consistent improvements on standard molecular and vision graph benchmarks, using sketch-based approximations (MinHash and HyperLogLog) for efficiency.

The main problem is that the contributions are overstated. The neighborhood intersection idea is already established in the literature, particularly in Chamberlain et al., where the same approximation techniques are used. The paper’s formulation adds little beyond integrating that encoding as a transformer bias. The technical development is inflated and overpormises.

**Strengths:**

Using k-hop commong neighborhoods as a proxy for structural information is an established technique in the literature. This paper proposes its application also as a structural encoding for graph transformer architectures, which is a natural and promising next step.

The proposed encoding has a simplicity to it that parallels prominent encodings such as RWSE or LapPE which can make it easily applicable for a wide variety of tasks.

The empirical results are very strong. The reported results are pushing the boundary on a variety of standard benchmarks, commonly surpassing elaborate state of the art architectures.

**Weaknesses:**

## Novelty / Attribution

The attribution to Chamberlain et al is insufficient. The idea of using pairwise encodings of k-hop neighborhoods as an efficient proxy for substructure information is the core of Chamberlain et al., which is already referenced but only in a vague sense regarding the methods to approimate these encodings. There, the formulation of “Following […], we propose […]” is not appropriate in my opinion. The approximation approach for pairwise k-hop common neighborhood is precisely present in Chamberlain et al., seemingly using the exact same techniques. In a way this work could even be summarized as using the techniques of Chamberlain et al. as a structural encoding for graph transformers. The one vague reference to their work is therefore  insufficient for proper attribution and recognition of novelty.

Given this large overlap with prior work, it remains unclear to me where the major novelty lies in the submission.

## Technical Development

The technical discussion in this paper is unfocused and unnecessarily opaque. Some concrete points towards this are listed below:

Line 191: why not just have the sum run for $x < k_u$ and $y < k_v$? Also it might be even simpler to define it via the neighborhoods, i.e., use that
$(N_{k_u}(u) \setminus N_{k_u-1}(u))$ are the edges exactly k_u steps from u and then take the intersection of this set with the corresponding set for v.

Line 198: similarly, just use $V(G) \setminus N_k(v)$ for the nodes that are further away than k from v and intersect with the precise k_u neighborhood as in the previous comment.

Line 205: given the previous comments, I find don’t understand the point in emphasizing the definition of $C_{k_u,k_v}$ since it seems to just complicate and obfuscate Definition 3.1 for no benefit.


Line 220: what is the point of the hash function on a single multiset? Theoretically you gain nothing since it’s just a bijection and in practice I think you use the actual multiset instead of the hash.


Theorem 3.3: the statements should be made more precise to avoid misleading the reader. Looking at the details in the appendix reveals that the results are much more narrow in scope than the theorem’s phrasing implies:
- It should be made clear that n here is the number of vertices because that does relativize points 1 and 2 quite a lot.
- In point 2 & 3,  “no less expressive” is very misleading. What is shown is that there exists one pair of graphs that can be distinguished by GIST(6) but not by RRWP(3) or RD-WL, nothing more.
- In point 3 of the theorem the some kS,kR is needlessly opaque. The proof is just for 3 and 6 specifically.
- In the proof you begin item 2 with a conjecture that seems unrelated to what is proven or what is stated in the theorem?

On top of this, SPD-WL and RD-WL are not defined and seemingly not even cited(??) in this submission.
I would have expected a statement on the relation to 2-FWL since that would be the most obvious point of comparison for representation on all node-pairs. It seems intuitive to me that 2-FWL actually is strictly more expressive than GIST and it would be helpful (and I believe of more interest than the RD-WL or RRWP results) to have a clear picture where to place GRIT in terms of expressivity.

Definition 4.1 is established standard and should be referenced. At the very least it should be made clear that this is not a contribution of the submission.

**Questions:**

Is there any reason that Theorem 4.3 is not trivial? Does this not follow immediately from the k-hop substructure encoding being invariant under isomorphism? If so clarify that this is the case when stating the theorem or drop it entirely. (Or maybe make it a proposition about the encoding?). If not, please make it clear in the paper why this theorem is needed.

Can you please clearly describe what the difference is between what you propose and what is already done in Chamberlain et al.? Given that can you please make the novelty of this submission clearer.

---

> ### Author Response · Authors · 2025-11-21
> **Rebuttal by Authors**
>
> ### **`W1, Q2 - Novelty of GIST` - It is non-trivial to apply prior structural features directly to Graph Transformer for graph classification task while GIST addresses a key limitation of prior methods: modeling higher-order structural relationships.**
> We thank the reviewer for pointing this out and agree that the attribution to Chamberlain et al. should be made more explicit. To clarify, we do **not** claim the efficient estimation of pairwise $k$-hop common-neighbor intersection as a contribution of our work; we fully acknowledge that this estimator originates from Chamberlain et al. Our intention in the original text was precisely to use their estimator as a building block. In the revised version, we have updated the wording to reflect this more clearly. We now explicitly state that we adopt the efficient estimator of Chamberlain et al. for computing pairwise $k$-hop common-neighbor cardinalities.
>
> However, we respectfully argue that although our method and Chamberlain et. al. may both utilize conceptually similar common neighbor statistics, the **intended purposes and applications are fundamentally different**. The approach of Chamberlain et. al. is specifically designed to address the **automorphic node identification problem**—that is, distinguishing structurally equivalent nodes in link prediction tasks. In contrast, our method leverages the cardinality of the common neighbor set as a pairwise feature to **capture higher-order structural relationships between node pairs**: a distinct and non-trivial challenge in Graph Transformers for graph-level tasks that, to the best of our knowledge, **we are the first one to propose**.
>
> Our proposed method, GIST, is explicitly designed with graph-level tasks in mind. While pairwise structural features have been shown to be effective for link prediction, as in \[1], we found that a **direct and vanilla application of this idea to graph-level tasks does not work**. To support this, we include a variant of a Graph Transformer that uses \[1] as a positional encoding baseline.
>
> Specifically, for each node $u \in V$, we define its positional encoding as:
>
> $$
> I_{k_u}(u) = \frac{1}{|V|}\sum_{v \in V} I_{k_u,k_v}(u,v)
> $$
>
> $$
> B_{k_u}(u) = \frac{1}{|V|}\sum_{v \in V} B_{k_u,>k}(u,v)
> $$
>
> $$
> PE(u) = [I_{k_u}(u) \, || \, B_{k_u}(u)]
> $$
>
> for every $k_u < k$, and add $PE(u)$ directly to the node embedding as a positional encoding. The result of this baseline is reported in Table 1.
>
> These findings empirically validate our claim: **successfully adapting common neighbor features to graph-level tasks requires more than direct reuse**. Thus, we believe that identifying and adapting common neighbor interactions for this purpose constitutes a **meaningful and novel contribution**. Furthermore, both Reviewer `o64B` and `idmr` agrees that our use of pairwise structural features provides a conceptually novel and meaningful structural bias—a strength, not a weakness.
>
> **We emphasize that the goal is to solve the problem, but not to solve in a particular way that might look good on paper.** While intersection-based features have been used in link prediction, their direct application to graph-level Graph Transformers has not been explored—and, as our experiments show, a naïve reuse does not work. Here we raise two questions:
>
> - *"Could we add extra modules to make the method look more sophisticated?"* Yes.
> - *"But should we do so?"* No—because it offers no real benefit and detracts from the practicality and adoption we aim to support in the community.
>
> **Criticizing our use of intersection features in this new context is akin to criticizing the use of random-walk ideas from PageRank when adapting them to Graph Transformers**: the conceptual tool may be similar, but the task, modeling objectives, and necessary adaptations are fundamentally different. Our findings—and the positive assessments from Reviewers `o64B` and `idmr`—confirm that our adaptation is non-trivial, necessary, and meaningfully advances structural modeling, validating the novelty and practical value of GIST. We hope these rationales help the reviewer revisit their assessment of novelty and recognize the genuine contribution our work makes to graph-level structural modeling.
>
> **Table 1**: Performance of GIST vs positional variant (GIST-PE)
> | Datasets | ZINC | Peptides-struct |
> |-|:-:|:-:|
> | GIST | **0.050** | **0.2442** |
> | GIST-PE | 0.102 | 0.2537 |

---

> ### Author Response · Authors · 2025-11-21
> **Rebuttal by Authors**
>
> ### **`W2 - Theoretical exposition of GIST` - We clarify misunderstood definitions and refine the expressivity discussion to better convey our theoretical intent.**
>
> We thank the reviewer for the careful reading of our theoretical section. Based on the feedback, we believe the reviewer’s suggestions and concerns can be categorized into *two main themes*:
>
> 1. Clarification of theoretical formulations and definitions
> 2. Scope of Theorem 3.3, and its relation to prior frameworks (RD-WL, SPD-WL, 2-FWL)
>
> Below, we address each in detail.
>
> ---
>
> #### **1. Clarification of theoretical formulations and definitions**
>
> * **L191**: We do not sum over $(x < k_u)$ and $(y < k_v)$ because $I$ measures the cardinality of neighborhoods that are *exactly* $(k_u, k_v)$-hops from $(u, v)$. This requires excluding $I_{k_u, k_v-1}$ and $I_{k_u-1, k_v}$, but not $I_{k_u, k_v}$ itself.
>
> * **L191 and L198**: While we agree that using explicit neighborhood sets (e.g., $N_{k_u}(u) \cap N_{k_v}(v)$) is a valid alternative formulation, our recursive 2D-matrix definition remains equivalent and is *chosen intentionally for clarity of implementation*. Since we cannot release code at this stage, we aim to provide a formulation that is easier for readers to follow and reproduce. Indeed, this effort has been appreciated by reviewer `idmr`, while reviewer `o64B` even asks for pseudo-code on implementation of GIST's computation.
>
> * **L205**: We thank the reviewer for noticing this. This sentence was intended to motivate the need for an efficient computation of $C_{k_u, k_v}(u, v)$ and to lead smoothly into Section 3.4. It was misplaced during page-limit edits, and we have corrected this in the revised version.
>
> * **L220**: Our intent is to present both a theoretical formulation (for establishing GIST’s expressiveness) and a practical formulation (for computing it efficiently). The notion of “coloring’’ nodes using hash values is standard in WL-style analyses and widely adopted in recent literature (e.g., [1][2]). We later build on this hashing formalism in Section 3.3. In contrast, Lines 224–228 introduce the practical definition of GIST to support readers from both theoretical and implementation-oriented backgrounds. We hope this clarifies the structure and resolves the confusion.
>
> * SPD-WL refers to the Shortest-Path-Distance Weisfeiler–Leman variant, and RD-WL refers to the Resistance-Distance WL variant used in reference [1] (cited at `L240`). Nevertheless, we agree that this might benefit from clearer definition and have added explicit explanations in the revised draft.
>
> Regarding the comment:
>
> > *“Definition 4.1 is established standard and should be referenced. At the very least it should be made clear that this is not a contribution of the submission,”*
>
> we would like to clarify that we **never intended** Definition 4.1 to be viewed as a contribution. We believe it is a standard notion of isomorphism-invariant node/edge colorings used widely in the WL and graph isomorphism literature, and we introduced it only for notation purpose.
>
>
>
> [1] Zhang et. al., Rethinking the ex- pressive power of GNNs via graph biconnectivity. In ICLR 2023.
>
> [2] Ma et. al., Graph inductive biases in transformers without message passing. In ICML 2023.

---

> ### Author Response · Authors · 2025-11-21
> **Rebuttal by Authors**
>
> ### **`W2 - Theoretical contribution of GIST(cont.)`**
>
> #### **2. Scope of Theorem 3.3, and relation to RD-WL, SPD-WL, and 2-FWL.**
> We thank the reviewer for the detailed feedback on Theorem 3.3 and its proof. We respectfully but firmly disagree with the concern that our theoretical part is one of the weaknesses. Our theorem and the proof explicitly shows that $\text{GIST}$ is strictly more expressive than shortest path distance (SPD) as a structural bias, and that there exist cases where $\text{GIST}$ can distinguish two graphs that resistance distance (RD-WL) or relative random walk positional encodings (RRWP) cannot. In our view, this level of comparison is a meaningful and sufficient theoretical contribution to support the main methodological contribution of introducing a new structural encoding.
>
>
> We agree that there can be different perspectives on how expressiveness results should be phrased, and we clarify our intent below.
>
> First, The reviewer asks `"Theorem 3.3...It should be made clear that n here is the number of vertices"`. We note that $n$ here is indeed the number of verticies, and we kindly refer the reviewer to `L251` where we clearly define $n$.
>
>
> Second, regarding the comment:
>
> > *“In point 2 & 3, ‘no less expressive’ is very misleading. What is shown is that there exists one pair of graphs that can be distinguished by GIST(6) but not by RRWP(3) or RD-WL, nothing more.”*
>
> We are sorry for the confusion. Our intended use of "no less expressive" is: GIST is neither strictly less expressive than nor strictly more expressive than RRWP or RD-WL. Theorem 3.3 gives a concrete pair of non-isomorphic graphs that $\text{GIST}$ can distinguish while $\text{RRWP}$ or $RD-WL$ cannot. We believe this comparison is sufficient to rule out the interpretation that GIST is strictly weaker; we do not claim any universal dominance or a full characterization of all graph pairs. We have updated the wording in the revised version to make our intention clearer.
>
> Third, the reviewer writes:
>
> > *“In point 3 of the theorem the some $k_S,k_R$ is needlessly opaque. The proof is just for 3 and 6 specifically.”*
>
> We are sorry for the confusing notation. The parameters $(k_S, k_R)$ were introduced only to mirror how we formalized the proof: we stated the result for $k_R < k_g$ so as not to over-claim beyond what is directly shown. In fact, for the specific example we construct, even taking $k_R = k_g$ would not help RRWP, since $\mathrm{RRWP}({k_g})$ still cannot distinguish the two graphs (it already fails at the truncated level). The purpose of point 3 is therefore purely to show that there exists a concrete pair of graphs that $\mathrm{GIST}$ can distinguish while $\mathrm{RRWP}$ cannot, and we have updated the wording to make this intent clearer.
>
>
>
> Next, the reviewer notes:
>
> > *“In the proof you begin item 2 with a conjecture that seems unrelated to what is proven or what is stated in the theorem?”*
>
> In item 2, our goal is to support the statement that in some cases, GIST is more expressive than RD-WL. The provided graph pair is drawn from a family of graphs that share the same effective-resistance spectrum. Prior works have widely observed that RD-WL cannot distinguish pairs of graphs in this family, whereas we show that $\mathrm{GIST}$ can. This directly underpins our claim about relative expressiveness. We use the word “conjecture” here to avoid over-claiming beyond what is formally proved.

---

> ### Author Response · Authors · 2025-11-21
> **Rebuttal by Authors**
>
> ### **`W2 - On the non-triviality of GIST's theoretical contribution.`**
>
> While we appreciate the reviewer’s suggestion to relate our expressiveness to $2$-FWL, we would like to clarify our focus. Our work is not intended as a full theoretical study of graph model expressiveness; rather, the theoretical results are designed to **support** the main methodological contribution of a new structural encoding. As discussed above, we already provide non-trivial expressiveness guarantees: $\text{GIST}$ is strictly more expressive than SPD-WL, and there exist graph pairs where $\text{GIST}$ distinguishes graphs that RD-WL or RRWP do not. This level of comparison is on par with, and in some respects stronger than, prior structural-encoding works.
>
> We would also like to put this in context with prior work. Concretely, GRIT [1] only proves that its structural encoding is more expressive than SPD, and SPSE [2] does not even establish strict expressiveness improvements over all prior structural encodings, but rather only provides constructed graph pairs where its encoding has advantages. By comparison, our analysis simultaneously (i) establishes strict expressiveness over SPD-WL and (ii) proves existence-based comparison against RD-WL and RRWP. From this perspective, we believe the scope and strength of our theoretical guarantees are at least on par with, and in some respects stronger than, these well-established works in the community.
>
> In contrast, $2$-FWL is widely regarded as a strong upper bound on pairwise expressiveness. Formally emphasizing that GIST is strictly less expressive than $2$-FWL would therefore amount to restating an unsurprising fact (shared with most practical encodings) without yielding actionable insight for Graph Transformer design. For a methodological paper, we believe it is more valuable to strike a balance: provide enough theory to substantiate the empirical claims and clarify where GIST stands relative to *practically used* baselines, while keeping the main focus on a scalable and effective structural bias rather than matching the strong known theoretical test.

---

> ### Author Response · Authors · 2025-11-21
> **Rebuttal by Authors**
>
> ### **`Q1 - Importance of Theorem 4.3` - We believe it is important to point this out for people not from graph theory background.**
> We appreciate the reviewer’s comment. Although the $k$-hop statistics used by GIST are isomorphism-invariant, we believe it is not necessarily obvious to all readers that this invariance is preserved once these features are introduced into the attention mechanism. Because a transformer must maintain permutation invariance at the attention level, we included Theorem 4.3 to make this property explicit. Indeed, reviewer `o64B` appreciate the explicit claim on the isomorphism-invariance of GIST.
>
> Attention-level invariance can be easy to break in practice. Architectural choices that appear natural—such as node-specific projections in queries/keys, or pair-dependent transforms $W_{u,v}$ applied to $S_k(u,v)$—may violate permutation invariance even when the underlying graphs are identical. Readers without background in permutation-equivariant design may reasonably assume such variants are safe when they are not.
>
> SPSE [3] provides an instructive example of this sensitivity. While the true simple-path tensor is invariant, the approximate SPSE encoder used in practice relies on DFS/BFS-based DAG constructions whose ordering and traversal decisions introduce asymmetries. Aggregating over only a finite set of such DAGs can yield structural biases that are not invariant.
>
> Given this, Theorem 4.3 is not meant as a major theoretical result, but as a verification step ensuring that the way GIST is integrated into attention does not inadvertently break invariance. In response to the reviewer’s suggestion, we have restated it as a proposition and move it to the appendix.
>
>
> [2] Graph Inductive Biases in Transformers without Message Passing, Ma et. al., ICML 2023.
>
> [3] Simple Path Structural Encoding for Graph Transformers, Airale et. al., ICML 2025.
>
> [4] Rethinking the expressive power of GNNs via graph biconnectivity, Zhang et. al., ICLR 2023.

---

> ### Author Response · Authors · 2025-11-25
> **A gentle reminder, and an invitation to discuss.**
>
> Dear reviewer `1jQu`,
>
> We would like to gently remind you to take a look at our rebuttal and, if possible, share any further thoughts, as addressing your concerns is very important to us.
>
> Your two main concerns were (1) the novelty of GIST and (2) its theoretical foundation. Regarding (1), we clarify that while the tool is similar to [1], the problem setting is fundamentally different, and we provide evidence that directly applying [1] to graph-level Graph Transformers is non-trivial. We also highlight that, to the best of our knowledge, **GIST is the first to explicitly target higher-order structural relationships between node pairs in Graph Transformers**. This is appreciated by Reviewers `o64B` and `idmr`.
>
> On the theory side, we have refined the wording around our expressiveness results. As our work is a methodological work, we kindly ask for the reviewer’s leniency and understanding in evaluating the scope and depth of our theoretical results, which we believe to be on par with, or even stronger than, these prior methodological works such as GRIT or SPSE.

---

> > ### Comment · Reviewer_1jQu · 2025-11-27
> >
> > Thank you very much for the reply and the extensive comments regarding my review.
> >
> > I have read the replies and the new version of the paper and I appreciate that the changes regarding the technical presentation have been mostly addressed and I have raised my score in terms of presentaiton and overall recommendation accordingly.
> >
> > However, I wish to highlight to the AC that my main concerns are misrepresented by this final rebuttal comment in the chain. My issue was not with the theoretical foundation but with (1) problematic/false overclaiming regarding the expressivity results, and with opaque technical presentation (this has been resolved), and (2) with, in my eyes, an inappropriate presentation of the scientific context that this work emerged from.
> >
> > As stated in my review, I find the principal idea of using the techniques of Chamberlain et al. as a structural encoding promising. However, even in the revision, presents the idea as if conceived by the authors, with a unassuming reference to Chamberlain et al.. In the first rebuttal comment this criticism is handwaved away by claiming that the "intended purposes [...] are fundamentally different". In my view, this misrepresents the discussed work since that work exactly uses MinHash and HyperLogLog as estimators to compute feature vectors that represent local structure around nodes in order to capture higer-order structural relationships.
> >
> > I believe the following quote of the first rebuttal comment illustrates my issues from another perspective
> >
> > >"Could we add extra modules to make the method look more sophisticated?" Yes.
> > > "But should we do so?" No
> >
> > I am saying the opposite. I don't think that you can add anything extra, because the modules you add are only exactly those that are implemtable with the method of Chamberlain et al.. And even that alone would be ok, if the paper would actually transparently present things that way, instead of first framing things slightly different on the surface (intersection signatures vs. subgraph sketching) before falling back on precisely the methods of prior work when it comes to practical implementation.

---

> ### Author Response · Authors · 2025-12-01
> **Thanks for your recognition, but we respectfully argue that GIST is addressing a new problem within graph-level tasks.**
>
> We first thank the reviewer for recognizing that using neighborhood intersection cardinality as a structural encoding is a promising direction, and for acknowledging that our revised version has addressed the earlier technical concerns. We believe the main remaining issue is the *“clearer positioning of GIST vs. Chamberlain et al.”*, which we would like to address faithfully here.
>
> While we agree that the paper benefits from a clearer positioning relative to Chamberlain et al., and have therefore added a dedicated discussion in the revised manuscript, we respectfully but firmly maintain our position that:
>
> > ## **The intended purposes and applications of the two methods are fundamentally different.**
>
> Please hear us out.
>
>
> ---
> `First, regarding your comment`:
>
> > "... that work exactly uses MinHash and HyperLogLog as estimators ... in order to capture higer-order structural relationships."
>
> We respectfully note that this claim does not appear to be supported by [1]. Chamberlain et al. **never claims to capture higher-order structural relationships; instead, its stated focus is on the automorphic node problem**. For example, at the end of page 2, the paper states:
>
> > "ELPH is strictly more expressive ... solves the automorphic node problem."
>
> Then Proposition 4.1 states:
>
> > "... ELPH does not suﬀer from the automorphic node problem."
>
> along with countless mentioning of *automorphic node* in the paper to highlight the main target of the method: **solving automorphic node problem of GNN**.
>
> By contrast, to the best of knowledge, we are **the first to explicitly proposes using intersection cardinality to capture higher-order structural relationships for Graph Transformers**. On `L60`, we states:
>
> > "... the first to promote aggregation across heterogeneous substructures by capturing higher-order relationships ..."
>
> Then on `L229-230`, we again state:
>
> > "GIST ... capturing higher-order relational dependencies among nodes and substructures."
>
> We also dedicate Appendix F to empirically demonstrating how GIST captures higher-order structural relationships through controlled experiments and case studies visualization.
>
> ---
>
> `Secondly, regarding your statement`:
>
> > “I don't think that you can add anything extra, because the modules you add are only exactly those that are implementable with the method of Chamberlain et al.”
>
> we firmly disagree with this claim. Beyond the configuration presented in the paper, we have explored several *more complicated* variants that performed competitively but were intentionally omitted to keep GIST simple, broadly applicable, and easy to adopt:
>
> 1. **Random-walk–powered GIST.** We experimented with applying GIST on each higher powers of the random-walk transition (i.e., $i$-th power graphs) to derive a stack of intersection features. This variant is more expressive but introduces extra complexity and overhead that we judged unnecessary for our core message.
>
> 2. **Multi-scale GIST via graph coarsening.** We also tested computing GIST on multiple coarsened versions of the graph to encode structural relations at different scales. While promising, this multi-level design significantly complicates implementation.
>
> 3. **GIST as structural/positional encoding.** As mentioned in our previous rebuttal, we explored injecting GIST directly as structural/positional encodings into Graph Transformer attention.
>
> We ultimately chose the current version of GIST to emphasize clarity, reproducibility, and widespread adoption, **not because the method is limited to what is “exactly implementable” by Chamberlain et al.** We see no reason to present a much more complicated variant when a simpler design already achieves our objectives and is more likely to be understood, implemented, and reused. We hope the reviewer shares this philosophy and agrees that, in this context, simplicity in the core design is a strength rather than a limitation.

---

> > ### Author Response · Authors · 2025-12-01
> > **Thanks!**
> >
> > Third, on the point of:
> >
> > > "first framing things slightly different on the surface (intersection signatures vs. subgraph sketching) before falling back on precisely the methods of prior work when it comes to practical implementation."
> >
> > We thank the reviewer for raising this important point. While we agree with the reviewer that directly adopting the terms in [1] such as "subgraph sketching" might position ourselves better to [1], we respectfully argue that:
> >
> > 1. **Different scope of what is encoded.** We want to highlight upfront that *we are not sketching any subgraphs like Chamberlain et al.* GIST is designed to encode the **entire graph structure and higher-order relationships**. In contrast, [1] **explicitly sketches localized subgraphs** to address the automorphic node problem. We hope the reviewer can agree with us that adopting subgraph sketching here is *even more misleading*.
> > 2. **Clarity of terminology.** To the best of our knowledge, `sketching` is a term heavily associated with hashing community. We therefore believe that directly foregrounding the term “intersection cardinality” (which is also a core underlying concept in [1]) makes the method more intuitive and accessible to the broader graph learning community.
> > 3. **Common Neighbors is well-known in IR.** Intersection-based measures, such as common neighbors, are standard in information retrieval and graph mining. Since GIST generalizes the idea of intersection-based structural signals, directly using **intersection-based terminology** is both natural and clearer than introducing sketching-specific jargon like [1].
> >
> > Nevertheless, we again agree with the reviewer that our paper would benefit from a clearer positioning vs. [1], especially since we adopt their estimators to efficiently capture the k-hop neighborhood intersection between node pairs. In the revised manuscript, we have therefore made the following changes:
> > 1. On `L182` where we define the $k$-hop intersection substructure between a pair of nodes, we explicitly states "following Chambelain et al., we encode..."
> > 2. On `L228-L229`, we highlight the difference between different objectives of [1] (subgraph sketching) vs. GIST (overall graph structure encoding with higher-order structual relationships).
> > 3. `L501-L508`, we add a dedicated paragraph to **faithfully highlight the  comparison of [1] vs. GIST**: what we adopt and what the difference between [1] and GIST is.
> >
> > ---
> >
> > We hope these updates ease the reviewer’s concerns about the positioning of our method relative to the prior work we adopt. We are extremely grateful that the reviewer finds our work interesting and views the use of intersection cardinality in Graph Transformers to target higher-order structural relationships as a promising direction.
> >
> > While we agree that the paper benefits from clearer positioning and have now addressed this faithfully in the revised manuscript, we believe this issue is largely **one of exposition and stylistic preference rather than a substantive limitation of the method**. Given the strong empirical performance and the interesting direction opened up by GIST, we kindly ask for the reviewer’s leniency and understanding, and hope that **our clarifications motivates another reconsideration of the overall rating**.
> >
> > [1] Graph Neural Networks for Link Prediction with Subgraph Sketching, Chamberlain et al., ICLR 2023.

---

### Official Review · Reviewer_idmr · 2025-10-30

**Soundness:** 3
**Presentation:** 3
**Contribution:** 3
**Rating:** 6
**Confidence:** 3

**Summary:**

This paper introduces GIST (Gisty Intersection Signature Trait), a novel structural encoding method for Graph Transformers. GIST encodes the intersection cardinalities of k-hop neighborhoods between node pairs, producing a permutation-invariant and expressive representation of graph substructures. The authors propose an efficient randomized estimation algorithm using MinHash and HyperLogLog, reducing computational complexity. GIST is integrated into the self-attention mechanism as an additional structural bias, enhancing the model’s capacity to capture fine-grained substructures and long-range dependencies.

Empirical evaluations across ZINC, Peptides, MoleculeNet, and LRGB benchmarks demonstrate consistent improvements over strong baselines such as GraphGPS, GRIT, Subgraphormer, and SPSE. Theoretical analysis shows that GIST is at least as expressive as several existing distance-based or random-walk-based positional encodings, and in some cases strictly more expressive.

**Strengths:**

1. The idea of using intersection cardinalities of k-hop neighborhoods is both intuitive and mathematically grounded. It provides a meaningful bridge between substructure-level and global structural information.
2. The paper offers a formal expressiveness comparison to other positional encodings (e.g., SPD-WL, RD-WL, RRWP). The proof sketch and theoretical guarantees add credibility and clarity to the method’s contribution.
3. The use of randomized hashing (MinHash + HyperLogLog) to approximate neighborhood intersections is elegant and practical. The computational reduction is well justified, and implementation details are transparent.

**Weaknesses:**

1. While GIST’s theoretical expressiveness is well discussed, the paper does not empirically isolate how much of the gain comes from the theoretical novelty versus implementation-level improvements (e.g., intersection vs. distance-based encoding).
2. Although the authors claim robustness, the main text lacks quantitative sensitivity analysis (delegated to the appendix). A concise presentation of trade-offs between accuracy and computational cost would strengthen the practical message.

**Questions:**

See weaknesses.

---

> ### Author Response · Authors · 2025-11-21
> **Rebuttal by Authors**
>
> ### **`W1 - Ablation study on importance of GIST` - Sure, here we provide additional ablations to quantify the effectiveness of GIST.**
>
> We thank the reviewer for this suggestion. We provide an ablation study comparing individual structural encodings (GRIT, RRWP, SPD)  to assess the effectiveness of GIST. We note that the attention architecture of GIST is the vanilla attention architecture, thus attention + SPD and attention + RRWP are essentially Graphormer and GRIT baselines.
>
> As shown in **Table 1**, **GIST alone outperforms other structural encodings**. While both GIST and RRWP capture structural information, our empirical results suggest that GIST serves as a stronger structural prior.
>
>
> **Table 1**: Performance of GIST vs. RRWP and SPD
> | Datasets | ZINC | Peptides-struct |
> |-|:-:|:-:|
> | SPD | 0.122 | _ |
> | GIST | 0.050 | 0.2442 |
> | RRWP | 0.059 | 0.2460 |
>
> ### **`W2 - Approximation error and performance tradeoffs` - Sure, we have moved the relevant ablations back to the main text.**
>
> We thank the reviewer for this suggestion. We originally placed this ablation study in the appendix due to page limitations, but we agree that highlighting the robustness and error–cost–performance tradeoff strengthens the practical relevance of our method. Accordingly, we have moved this section back into the main text in the updated version.
> Table 4, which varies the number of $k$-hop layers, demonstrates that GIST remains robust across different choices of $k$. Tables 5 and 6 vary the number of MinHash functions and the $p$ parameter of HyperLogLog. Theoretically, increasing these values reduces the estimation error of GIST, while in practice it slightly increases computational overhead. Across these settings, we observe a clear trade-off between efficiency and performance, and GIST remains fairly robust throughout.

---

> > ### Comment · Reviewer_idmr · 2025-11-25
> >
> > Thank you for the response. It solves my concerns. I'd like to keep my current positive score.

---

### Official Review · Reviewer_o64B · 2025-10-31

**Soundness:** 3
**Presentation:** 3
**Contribution:** 3
**Rating:** 8
**Confidence:** 4

**Summary:**

The paper “Get the GIST of Graphs with Intersection Signature (GIST)” proposes a novel structural encoding method for Graph Transformers, addressing their well-known difficulty in effectively capturing graph topology.Traditional Graph Neural Networks (GNNs) suffer from oversmoothing and oversquashing due to local message passing, while Graph Transformers, though capable of modeling long-range dependencies, rely heavily on how structural information is encoded. Existing encodings (e.g., shortest path, Laplacian, or random walk features) only partially represent graph structure and often fail to model interactions among diverse substructures.

To overcome this, the authors introduce GIST (Gisty Intersection Signature Trait) : a structural encoding based on the intersection cardinalities of k-hop neighborhoods between node pairs. GIST serves as a permutation-invariant and expressive representation that captures fine-grained substructure interactions across the entire graph.

The main contributions are:

Novel Encoding Framework: GIST encodes graph structure using pairwise intersection features of k-hop neighborhoods, effectively capturing both local and global substructure relationships.

Efficient and Scalable Computation: The authors design a randomized estimation method (using MinHash and HyperLogLog) to compute GIST efficiently, reducing complexity from O(k2n4)

Integration with Graph Transformers: GIST is incorporated as a learnable attention bias (“GIST attention”), allowing Transformers to use these structural cues for improved node pair interactions. The method is proven to be isomorphism-invariant and theoretically more expressive than several existing structural encodings.

Comprehensive Evaluation: Extensive experiments on 14 benchmark datasets (including ZINC, MoleculeNet, Peptides, MNIST, and CIFAR10) show that GIST consistently achieves state-of-the-art or near state-of-the-art performance. It also demonstrates strong scalability, robustness to hyperparameters, and effective generalization to node- and graph-level tasks.

Overall, the paper contributes a simple, theoretically sound, and empirically effective structural encoding that strengthens Graph Transformers’ ability to model complex substructures and long-range dependencies in graphs.

**Strengths:**

Originality:
The paper introduces a highly original structural encoding method, GIST (Gisty Intersection Signature Trait), that reframes how Graph Transformers capture graph topology. Unlike prior positional or distance based encodings, GIST leverages the intersection cardinalities of k hop neighborhoods to characterize higher order substructure interactions. This formulation is both conceptually novel and technically elegant, offering a new way to integrate relational information without relying on handcrafted motifs or message passing. The randomized estimation approach using MinHash and HyperLogLog further demonstrates creative thinking in scaling theoretical insights to practical use.

Quality:
The technical quality of the paper is strong. The authors provide solid theoretical grounding, showing that GIST is isomorphism invariant and strictly more expressive than several established encodings such as SPD WL and RRWP. The experimental design is thorough, covering 14 benchmark datasets across molecular property prediction, long range dependency modeling, and graph classification. Comparisons with a comprehensive set of strong baselines including GraphGPS, GRIT, and Subgraphormer reinforce the robustness of results. Implementation details, ablation studies, and scalability analyses are clearly reported, adding confidence in reproducibility and validity.

Clarity:
The paper is well organized and generally easy to follow despite the technical depth. The motivation is clearly articulated, with intuitive figures such as substructure intersection visualizations that help convey the core idea. Definitions are rigorous yet readable, and the progression from theoretical formulation to algorithmic implementation is smooth. The writing effectively balances formalism and intuition, which makes the contribution accessible to both theoretical and applied audiences in graph learning.

Significance:
The work makes a substantive contribution to the design of structure aware Graph Transformers, a rapidly growing research area. By providing a scalable, expressive, and permutation invariant encoding, GIST has clear potential to become a standard component in future graph Transformer architectures. Its ability to model complex substructure interactions extends the reach of Transformers to tasks in chemistry, biology, and materials science, where structural detail is crucial. The simplicity and generality of GIST also make it a promising foundation for future work in efficient graph representation learning and model interpretability.

**Weaknesses:**

Limited conceptual comparison to closely related structural encodings:
While GIST presents a novel formulation based on intersection cardinalities, the paper could strengthen its conceptual positioning relative to recent graph Transformer encodings that also aim to capture higher order or substructure information. For example, works such as SPSE (Airale et al., 2025) and FragNet (Wollschlager et al., 2024) introduce path and fragment based encodings that similarly emphasize richer structural dependencies. The paper briefly cites these methods but does not provide a deeper analytical or empirical comparison to clarify what unique structural biases GIST captures beyond them. A more systematic ablation comparing GIST against these structurally similar encodings under matched model capacity would make the originality claim more robust.

Theoretical exposition could be expanded for clarity and intuition:
Although the authors provide a theoretical proof of expressiveness, the connection between the formal results (e.g., comparisons with SPD WL and RRWP) and the empirical improvements remains somewhat abstract. The theory section would benefit from a more intuitive explanation or illustrative examples that show how intersection cardinalities encode structural distinctions that distance based or random walk encodings miss. This would help bridge the gap between theoretical guarantees and the observed empirical performance.

Experimental scope could better test generalization and scalability claims:
The experimental results are comprehensive, but most benchmarks are medium scale molecular or synthetic datasets. To convincingly support the claim of scalability to large graphs, additional experiments on large scale graph datasets such as OGBN Papers100M or OGBG Code2 would be valuable. These would more concretely demonstrate GIST’s computational advantages from the MinHash and HyperLogLog estimations and clarify the tradeoff between approximation accuracy and model performance.

Limited ablation on key design components:
While the paper includes some ablations, they focus primarily on hyperparameters (e.g., hop count or hash functions). It would strengthen the work to include controlled analyses that isolate the contribution of the intersection based features themselves. For instance, comparing GIST attention against versions using only shortest path distance or neighborhood overlap without full k hop intersection vectors would more precisely quantify the gain from the proposed encoding. Similarly, evaluating the impact of randomized estimation error on downstream accuracy would support the efficiency claims more convincingly.

Presentation and reproducibility aspects:
Although the paper is generally well written, some definitions (e.g., the construction of k hop substructure vectors) are dense and may hinder reproducibility for readers less familiar with the notation. Providing pseudocode or a high level algorithm box summarizing the computation of GIST and its integration into attention would improve clarity. In addition, since the code release is deferred, the paper would benefit from including at least partial implementation details (e.g., hash function parameters, runtime complexity benchmarks) to ensure results can be independently verified.

**Questions:**

Clarification of theoretical expressiveness:
The paper claims that GIST is strictly more expressive than SPD WL and comparable to or stronger than RRWP and RD WL encodings. Could the authors provide an intuitive explanation or illustrative example showing how intersection cardinalities capture structural information that distance or random walk based encodings cannot? This would help readers connect the theory with the observed empirical gains.

Quantifying approximation error and scalability tradeoffs:
The proposed efficiency improvement relies on MinHash and HyperLogLog estimations. How much variance or bias do these approximations introduce in practice, and how does this affect model accuracy? Reporting runtime and accuracy tradeoffs, or error bounds on representative datasets, would make the scalability claims more convincing.

Comparison to recent structural encodings:
Recent approaches such as SPSE (Airale et al., 2025) and FragNet (Wollschlager et al., 2024) also capture substructure relationships. Could the authors clarify how GIST’s intersection based features differ conceptually or empirically from these methods? A targeted comparison under similar model settings would help isolate GIST’s unique contribution.

Generality and applicability beyond molecular datasets:
Most experiments focus on molecular graphs, which have regular and chemically meaningful structures. How would GIST perform on graphs with very different properties, such as citation or social networks? Discussing whether intersection based encoding remains effective in sparse, irregular, or high degree graphs would strengthen the generality claims.

---

> ### Author Response · Authors · 2025-11-21
> **Rebuttal by Authors**
>
> We thank the reviewer for acknowledging the strengths of our work and for providing constructive suggestions to improve the paper. Below, we offer clarifications to address the raised concerns.
>
> ---
>
> ### **`W1, Q3 – Comparison to closely related structural encodings` – Both suggested works are thoroughly discussed in the appendix and empirically compared in our experiments.**
>
> We believe the reviewer may have missed some details, so we kindly refer to Appendix B, where we provide a conceptual comparison between our method and both SPSE and FragNet. FragNet also plays a central role in motivating our formulation in Section 2. Importantly, both SPSE and FragNet are included as baselines in our empirical evaluations. We will consider moving the detailed `Related Works` section back to the main text in the updated version for better visibility.
>
> ---
>
> ### **`W2, Q1 – Theory clarity and intuition` – Additional example is provided in the appendix.**
>
> We kindly refer the reviewer to Appendix B, where we present a formal analysis of GIST’s expressiveness and include a pair of graphs that GIST can distinguish but SPD and RW cannot. This pair is carefully constructed from the same graph family and shares the same effective-resistance distance spectrum. This example concretely illustrates that GIST is strictly more expressive than these baselines in certain non-trivial cases. we hope this would help bridge the gap between theoretical guarantees and the observed empirical performance.
>
> ---
>
> ### **`W3 – Experiment generalization and scalability` – Additional large-scale results are provided.**
>
> We refer the reviewer to Appendix G for results on PCQM4Mv2 (~3.7M graphs), one of the largest graph-level benchmarks available. This dataset is substantially larger than the reviewer’s suggestion of OGBG-Code2, and we hope this helps alleviate concerns regarding scalability. Regarding OGBN-Papers100M, it is a node-level benchmark and falls outside the scope of the graph-level tasks addressed in our paper. Nonetheless, we appreciate the suggestion and will consider including large-scale node-level evaluations in future work.
>
> ---
>
> ### **`W4 – Ablation on key design components` – Sure, here we provide additional ablations to quantify the effectiveness of GIST.**
>
> We thank the reviewer for this suggestion. We provide an ablation study comparing individual structural encodings (GRIT, RRWP, SPD)  to assess the effectiveness of GIST. We note that the attention architecture of GIST is the vanilla attention architecture, thus attention + SPD and attention + RRWP are essentially Graphormer and GRIT baselines.
>
> As shown in **Table 1**, **GIST alone outperforms other structural encodings**. While both GIST and RRWP capture structural information, our empirical results suggest that GIST serves as a stronger structural prior.
>
>
> **Table 1**: Performance of GIST vs. RRWP and SPD
> | Datasets | ZINC | Peptides-struct |
> |-|:-:|:-:|
> | SPD | 0.122 | _ |
> | GIST | 0.050 | 0.2442 |
> | RRWP | 0.059 | 0.2460 |
>
>
> We also refer the reviewer to Tables 12 and 13, where we vary the number of MinHash functions and the $p$-value of HyperLogLog. Increasing these hyperparameters theoretically reduces the estimation error of the intersection cardinality while marginally increasing computation. These results show how randomized estimation error impacts downstream accuracy.

---

> ### Author Response · Authors · 2025-11-21
> **Rebuttal by Authors**
>
> ### **`W5 – Pseudocode or a high-level algorithm` – Sure, a detailed pseudo-code is provided.**
>
> We kindly refer the reviewer to Appendix D, which already contains detailed pseudocode for estimation algorithm, and we have updated with pseudo-code on computing GIST. Combined with the attention architecture in Section 4, we believe the current description is sufficiently clear for reproducibility. We are also working on obtaining permission to release the source code and hope to make it publicly available soon.
>
>
> ### **`Q2 - Approximation error and scalability tradeoffs` - Sure, please check appendix C.3.**
>
> We kindly refer the reviewer to Appendix C.3, where we provide a theoretical bound on the bias and variance of GIST’s estimation. From a practical perspective, we refer the reviewer to Tables 4, 5, and 6, where we vary the number of MinHash functions and the $p$-value of HyperLogLog. These hyperparameters control the trade-off between estimation accuracy and computational efficiency, and the results illustrate how GIST’s performance remains robust under different accuracy–efficiency-estimation error configurations.
>
> ### **`Q4 – Generality and applicability beyond molecular datasets` – Additional non-molecular results are provided.**
>
> We thank the reviewer for raising this point. We kindly refer to **MNIST** and **CIFAR10** in Table 1—both of which are *not* chemistry-related. Similarly, the **Pattern** and **Cluster** datasets in Table 10 are also outside the molecular domain. Across all of these non-molecular benchmarks, GIST maintains competitive performance, demonstrating that its effectiveness is not limited to molecular graphs. We hope these results help alleviate concerns regarding the generality and applicability of GIST beyond molecular datasets.

---

### Official Review · Reviewer_fbzd · 2025-10-31

**Soundness:** 2
**Presentation:** 2
**Contribution:** 2
**Rating:** 2
**Confidence:** 4

**Summary:**

This work introduces GIST (Gisty Intersection Signature Trait), which is a structural encoding for graph transformers that is based on the intersection cardinalities of k-hop neighborhoods between node pairs. The paper provides a set of theoretical expressivity claims for GIST and also conducts experiments to support the GIST’s practical utility.

**Strengths:**

- The motivation of incorporating neighbourhood intersection information as a structural encoding is sound and well laid out.
- Empirical results are quite strong, and the authors show consistent performance gains across several different datasets and domains.

**Weaknesses:**

- My main concern is with the novelty of this work. It seems like the GIST encoding/features (computed with MinHash and HyperLogLog) are exactly the same as those presented in Chamberlain et al.
    - How does GIST meaningfully build on the work presented in Chamberlain et al.? It seems like the authors simply took the features presented in that work and applied them to graph transformers here, which is not enough of a contribution to the machine learning field.
- The stated theoretical contributions don’t seem very well defined or supported. In particular:
    - RD-WL (as mentioned in statement 2 of Theorem 3.3) is not defined or cited anywhere. What exactly is this?
        - Furthermore, the authors only provide a single example for this, which does not constitute a rigorous proof. Can this statement be strengthened or formalised?
    - The proof for statement 3 of Theorem 3.3 also only provides a single example, and “some values of $k_R$ < $k_g$” is extremely vague. Do the authors have any more specific results for conditions under which this holds (ie for what $k_R$ and/or $k_g$)?
    * Theorem 4.3 seems quite trivial, and I’m not sure it’s notable enough to be considered a significant theoretical contribution.

Overall, this paper doesn’t seem fully fleshed out yet, and might be more suitable for a workshop submission in its current state (owing to the strong empirical results). Therefore, I recommend it for rejection.

References:

Chamberlain et al. Graph Neural Networks for Link Prediction with Subgraph Sketching. In ICLR 2023.

**Questions:**

See weaknesses.

---

> ### Author Response · Authors · 2025-11-21
> **Rebuttal by Authors**
>
> ### **`W1 - Novelty of GIST` - It is non-trivial to apply prior structural features directly to Graph Transformer for graph classification task while GIST addresses a key limitation of prior methods: modeling higher-order structural relationships.**
>
> We respectfully argue that although our method and \[1] may both utilize conceptually similar common neighbor statistics, the **intended purposes and applications are fundamentally different**. The approach in \[1] is specifically designed to address the **automorphic node identification problem**—that is, distinguishing structurally equivalent nodes in link prediction tasks. In contrast, our method leverages the cardinality of the common neighbor set as a pairwise feature to **capture higher-order structural relationships between node pairs**: a distinct and non-trivial challenge in Graph Transformers for graph-level tasks that, to the best of our knowledge, **we are the first one to propose**.
>
> Our proposed method, GIST, is explicitly designed with graph-level tasks in mind. While pairwise structural features have been shown to be effective for link prediction, as in \[1], we found that a **direct and vanilla application of this idea to graph-level tasks does not work**. To support this, we include a variant of a Graph Transformer that uses \[1] as a positional encoding baseline.
>
> Specifically, for each node $u \in V$, we define its positional encoding as:
>
> $$
> I_{k_u}(u) = \frac{1}{|V|}\sum_{v \in V} I_{k_u,k_v}(u,v)
> $$
>
> $$
> B_{k_u}(u) = \frac{1}{|V|}\sum_{v \in V} B_{k_u,>k}(u,v)
> $$
>
> $$
> PE(u) = [I_{k_u}(u) \, || \, B_{k_u}(u)]
> $$
>
> for every $k_u < k$, and add $PE(u)$ directly to the node embedding as a positional encoding. The result of this baseline is reported in Table 1.
>
> These findings empirically validate our claim: **successfully adapting common neighbor features to graph-level tasks requires more than direct reuse**. Thus, we believe that identifying and adapting common neighbor interactions for this purpose constitutes a meaningful and novel contribution. Furthermore, **both Reviewer `o64B` and `idmr`** agrees that our use of pairwise structural features provides a conceptually novel and meaningful structural bias—a strength, not a weakness.
>
> **We emphasize that the goal is to solve the problem, but not to solve in a particular way that might look good on paper.** While intersection-based features have been used in link prediction, their direct application to graph-level Graph Transformers has not been explored—and, as our experiments show, a naïve reuse does not work. Here we raise two questions:
>
> - *"Could we add extra modules to make the method look more sophisticated?"* Yes.
> - *"But should we do so?"* No, because it offers no real benefit and detracts from the practicality and adoption we aim to support in the community.
>
> **Criticizing our use of intersection features in this new context is akin to criticizing the use of random-walk ideas from PageRank when adapting them to Graph Transformers**: the conceptual tool may be similar, but the task, modeling objectives, and necessary adaptations are fundamentally different. Our findings—and the positive assessments from Reviewers `o64B` and `idmr`—confirm that our adaptation is non-trivial, necessary, and meaningfully advances structural modeling, validating the novelty and practical value of GIST. We hope these rationales help the reviewer revisit their assessment of novelty and recognize the genuine contribution our work makes to graph-level structural modeling.
>
> **Table 1**: Performance of GIST vs positional variant (GIST-PE)
> | Datasets | ZINC | Peptides-struct |
> |-|:-:|:-:|
> | GIST | **0.050** | **0.2442** |
> | GIST-PE | 0.102 | 0.2537 |
>
> [1] Graph Neural Networks for Link Prediction with Subgraph Sketching, Chamberlain et al., ICLR 2023.

---

> ### Author Response · Authors · 2025-11-21
> **Rebuttal by Authors**
>
> ### **`W2 - Theoretical foundation clarification` - We have clarified confusing points, and our supporting theoretical claims are on par with, or in some respects stronger than, those in prior methodological works.**
>
>
> We thank the reviewer for the detailed feedback on Theorem 3.3 and its proof. We respectfully but firmly disagree with the concern that our theoretical part is one of the weaknesses. Our theorem explicitly shows that $\text{GIST}(n-1)$ is strictly more expressive than shortest path distance (SPD) as a structural bias, and that there exist cases where $\text{GIST}$ can distinguish two graphs that resistance distance (RD-WL) or relative random walk positional encodings (RRWP) cannot. In our view, this level of comparison is a meaningful and sufficient theoretical contribution to support the main methodological contribution of introducing a new structural encoding.
>
> We would also like to put this in context with prior work. Concretely, GRIT [2] only proves that its structural encoding is more expressive than SPD, and SPSE [3] does not even establish strict expressiveness improvements over all prior structural encodings, but rather only provides constructed graph pairs where its encoding has advantages. By comparison, our analysis simultaneously (i) establishes strict expressiveness over SPD and (ii) proves existence-based comparison against RD and RRWP. From this perspective, we believe the scope and strength of our theoretical guarantees are at least on par with, and in some respects stronger than, these established works in the community. Within that context, we hereby provide clarification on the reviewer's concerns.
>
> ---
>
> ### 1. Clarifying SPD-WL and RD-WL
>
> SPD-WL refers to the Shortest-Path-Distance Weisfeiler–Leman variant, and RD-WL refers to the Resistance-Distance WL variant used in [4] (cited at `L240`). We agree that might benefit from clearer stated and have added explicit explanations in the revised draft.
>
> ---
>
> ### 2. Regarding Theorem 3.3 (Statements 2 and 3)
>
> We are sorry for the confusion. Our intention is that statement 3 should be interpreted as an existence statement, rather than an attempt to fully characterize all relationships between $\mathrm{GIST}(k_g)$ and $\mathrm{RRWP}(k_R)$. The goal is to show that there exist graph instances for which $\mathrm{GIST}(k_g)$ distinguishes graphs that $\mathrm{RRWP}(k_R)$ does not. The condition $k_R < k_g$ in the theorem is used only to avoid over-claiming, since our formal proof is written in that setting (with truncated RRWP); in fact, for the specific example we construct, even taking $k_R = k_g$ does not help RRWP, because $\mathrm{RRWP}({k_g})$ still cannot distinguish the two graphs (it already fails at the truncated level).
>
> More broadly, comparing the expressiveness of $\mathrm{GIST}$ and $\mathrm{RRWP}$ in full generality is inherently difficult. $\mathrm{GIST}(k)$ is driven by multi-hop neighborhood intersection patterns, while $\mathrm{RRWP}(k)$ is governed by $k$-step random-walk behavior (return probabilities, multi-step walk statistics, etc.). These encode different types of structural information, and their relative power depends on how intersection patterns and random-walk distributions interact on a given graph. Providing a complete expressiveness comparison between the two would therefore require a much deeper, dedicated theoretical analysis that goes beyond the scope of this methodological work. We have refined the wording in the paper to make this intention clearer.

---

> ### Author Response · Authors · 2025-11-21
> **Rebuttal by Authors**
>
> ### **`W2 - Theoretical foundation clarification (Cont.)`**
>
> ### 3. On the Non-Triviality of GIST’s Invariance (Theorem 4.3)
>
> We appreciate the reviewer’s comment. Although the $k$-hop statistics used by GIST are isomorphism-invariant, we believe it is not necessarily obvious to all readers that this invariance is preserved once these features are introduced into the attention mechanism. Because a transformer must maintain permutation invariance at the attention level, we included Theorem 4.3 to make this property explicit. Indeed, reviewer `o64B` appreciate the claim on the isomorphism-invariance of GIST.
>
> Attention-level invariance can be easy to break in practice. Architectural choices that appear natural—such as node-specific projections in queries/keys, or pair-dependent transforms $W_{u,v}$ applied to $S_k(u,v)$—may violate permutation invariance even when the underlying graphs are identical. Readers without background in permutation-equivariant design may reasonably assume such variants are safe when they are not.
>
> SPSE [3] provides an instructive example of this sensitivity. While the true simple-path tensor is invariant, the approximate SPSE encoder used in practice relies on DFS/BFS-based DAG constructions whose ordering and traversal decisions introduce asymmetries. Aggregating over only a finite set of such DAGs can yield structural biases that are not invariant.
>
> Given this, Theorem 4.3 is not meant as a major theoretical result, but as a verification step ensuring that the way GIST is integrated into attention does not inadvertently break invariance. In response to the reviewer’s suggestion, we have restated it as a proposition and move it to the appendix.
>
>
> [2] Graph Inductive Biases in Transformers without Message Passing, Ma et. al., ICML 2023.
>
> [3] Simple Path Structural Encoding for Graph Transformers, Airale et. al., ICML 2025.
>
> [4] Rethinking the expressive power of GNNs via graph biconnectivity, Zhang et. al., ICLR 2023.

---

> ### Author Response · Authors · 2025-11-25
> **A gentle reminder, and an invitation to discuss.**
>
> Dear reviewer `fbzd`,
>
> We would like to gently remind you to take a look at our rebuttal and, if possible, share any further thoughts, as addressing your concerns is very important to us.
>
> Your two main concerns were (1) the novelty of GIST and (2) its theoretical foundation. Regarding (1), we clarify that while the tool is similar to [1], the problem setting is fundamentally different, and we provide evidence that directly applying [1] to graph-level Graph Transformers is non-trivial. We also highlight that, to the best of our knowledge, **GIST is the first to explicitly target higher-order structural relationships between node pairs in Graph Transformers**. This is appreciated by Reviewers `o64B` and `idmr`.
>
> On the theory side, we have refined the wording around our expressiveness results. As our work is a methodological work, we kindly ask for the reviewer’s leniency and understanding in evaluating the scope and depth of our theoretical results, which we believe to be on par with, or even stronger than, these prior methodological works such as GRIT or SPSE.

---

> ### Comment · Reviewer_fbzd · 2025-11-28
>
> I thank the authors for their detailed response, which has addressed many of my concerns. I believe the claims in the theoretical contributions section are clearer and more reasonable now.
>
> I also wish to clarify that my comment on novelty was not to suggest that the authors should unnecessarily introduce a more sophisticated approach. Rather, if they are to repurpose established encodings techniques (such as using HyperLogLog and MinHash to estimate the cardinality of intersecting neighborhoods from Chamberlain et al.), there should be additional meaningful contributions on top of that.
>
> Although I am still not fully convinced that is achieved in this work given the updated limited scope on the theoretical side, the additional experiments with GIST-PE are an encouraging start in that direction, and I will increase my score to a 4 to reflect this.

---

> > ### Author Response · Authors · 2025-12-01
> > **Thanks for your recognition, but we respectfully argue that GIST indeed makes meaningful contribution.**
> >
> > We first thank the reviewer for acknowledging that our revised version has addressed the earlier technical concerns. We believe the main remaining issue is only the *“novelty of GIST”*, which we would like to address here.
> >
> > We would first like to note that, aside from the reviewer, the **other three reviewers all found our contribution meaningful and interesting**:
> >
> > - R `o64B`: *“This formulation is both conceptually novel and technically elegant.”*
> > - R `idmr`: *“It provides a meaningful bridge between substructure-level and global structural information.”*
> > - R `1jQu`: *“This paper proposes its application also as a structural encoding for graph transformer architectures, which is a natural and promising next step.”*
> >
> > Importantly, R `1jQu`, who raised concerns broadly similar to yours, still concluded after the rebuttal that *“I find the principal idea of using the techniques of Chamberlain et al. as a structural encoding promising.”*
> >
> > ---
> >
> > While we fully acknowledge and respect that different reviewers may have different standpoints on novelty or meaningfulness, we would like to offer a few hypothetical Q&As to articulate our perspective. We hope the reviewer can hear us out:
> >
> > 1. **`Is there any prior work that adopts well-known concepts and still has high impact in Graph Transformers?` Yes - GRIT is a prominent example.**
> >
> >     * GRIT directly adopts random walk features, which had already been extensively used in Graph Transformers (e.g., RWPE [1], GraphGPS [2]). Yet GRIT is widely viewed as one of the most impactful recent works in the area.
> >
> >     * In contrast, GIST (i) **is the first to** adopt intersection cardinality in Graph Transformers and (ii) leverages it specifically to **capture higher-order structural relationships**—an aspect that has not been addressed by prior work.
> >
> > 2. **`Is there a meaningful difference between Chamberlain et al. and GIST? Yes, local subgraph sketching vs. global structural relationships.`**
> >
> >    * Chamberlain et al. sketch *local* subgraphs to address the automorphic node problem. In contrast, GIST leverages intersection cardinality to capture *global* graph structure and its higher-order structural relationships. The estimators overlap, but **the object, purpose, and task are fundamentally different.**
> >
> >    * We also dedicate Appendix F to empirically demonstrating how GIST captures higher-order structural relationships through controlled experiments and case studies visualization.
> >
> > 3. **`Is the theoretical contribution of GIST limited?`  No.**
> >
> > 	* We respectfully but firmly disagree your claim that our revised version has a “limited scope on the theoretical side”. We risk of being redundant to state again our theoretical results are, in our view, **on par with—or even stronger than—other methodological works** such as GRIT or SPSE.
> >
> > ---
> >
> > Given all of these, we want to finally answer the question:
> >
> > > ## **Does GIST make meaningful contribution to the field of Graph Transformer?**
> >
> > We gladly answer **yes**—and we believe *the other three reviewers would agree this time*. GIST introduces a new way to encode higher-order structural relationships across the entire graph, integrated into Graph Transformer. These aspects **are not solved by Chamberlain et al.’s local subgraph sketches, nor by prior encodings such RWPE, RRWP, or SPSE**. We show that GIST captures structural dependencies that existing methods cannot, leading to consistent performance gains.
> >
> > With this in mind, we respectfully invite the reviewer to reconsider what should count as a “meaningful contribution” in this methodological space and **whether our work might warrant a higher rating**. If GRIT—based on long-established random-walk ideas—is viewed as impactful for recontextualizing a known concept, then we believe GIST provides an equally meaningful advancement by introducing a new structural encoding with clear theoretical and empirical benefits. If the reviewer feels this still does not qualify, we would appreciate clarity on what *would* constitute meaningful contribution for works of this nature.
> >
> > Warm regards,
> >
> > *Paper13433* Authors
> >
> >
> > [1] Graph Neural Networks with Learnable Structural and Positional Representations, Dwivedi et al., ICLR 2021.
> >
> > [2] Recipe for a General, Powerful, Scalable Graph Transformer, Rampasek et al., NeurIPS 2022.

---

### Author Response · Authors · 2025-12-01
**Post-rebuttal Summary of Reviewers' Feedback and Our Rebuttals**

We understand that navigating mixed reviews is always a muddy experience, regardless of how responsible an AC strives to be. Thus, here we provide a summary of our work, the feedback we received, and our rebuttals for convenient navigation.

We would also like to note that both reviewers who initially gave an overall score of 2 have **raised their ratings** after the first rebuttal, acknowledging that the main technical issues **were satisfactorily addressed**. Together with the two stronger reviews, this now places the paper at least at `8644`, which we hope signals that the work is moving in a positive direction.

---

## **TL;DR of Our Work**
We propose a novel structural encoding method for graph classification tasks based on pairwise $k$-hop neighborhood intersection between nodes. While prior graph transformer methods primarily focus on substructure similarity, to the best of our knowledge, we are **the first to advocate for aggregating representations across diverse substructures**. Our method demonstrates strong and consistent improvements over prior baselines across a wide range of benchmarks.

In short, we believe our contributions can be summarized as follows:
- The **first to advocate for aggregation across diverse substructures via structural feature** in Graph Transformers
- A **novel structural encoding that captures higher-order structural relationships** based on $k$-hop neighborhood intersection
- A **theoretical foundation** characterizing the expressiveness of GIST
- **Rigorous benchmarking and evaluation** across a wide range of baselines and datasets

___

## **Reviewers' Recognition**
Even though reviewer ratings are mixed, they recognize our work for its **strong performance & rigorous benchmarking**, **solid theoretical foundation**, **sound motivation**, and **novel contributions**:

- **Reviewers acknowledge the strong performance of our method, supported by rigorous benchmarking across diverse datasets:**
	- `fbzd`:  *"Empirical results are quite strong, ... consistent performance gains across several different datasets and domains."*
	- `o64B`:  *"Extensive experiments on 14 benchmark datasets ... show that GIST consistently achieves state-of-the-art or near state-of-the-art performance."*
	- `idmr`:  *"Empirical evaluations across ZINC, Peptides, MoleculeNet, and LRGB benchmarks demonstrate consistent improvements over strong baselines ..."*
	- `1jQu`:  *"The empirical results are very strong. The reported results are pushing the boundary on a variety of standard benchmarks, commonly surpassing elaborate state of the art architectures."*
- **Reviewers find GIST has a solid theoretical grounding for its expressiveness:**
	- `o64B`:  *"The authors provide solid theoretical grounding, showing that GIST is isomorphism invariant and strictly more expressive..."*
	- `idmr`:  *"The paper offers a formal expressiveness comparison to other positional encodings (e.g., SPD-WL, RD-WL, RRWP). The proof sketch and theoretical guarantees add credibility and clarity to the method’s contribution."*
- **Reviewers consistently praise the clarity, intuition, and sound motivation of GIST’s design:**
	- `fbzd`:  *"The motivation of incorporating neighbourhood intersection information as a structural encoding is sound and well laid out."*
	- `064B`:  *"The motivation is clearly articulated ..."*
	- `idmr`:  *"The idea of using intersection cardinalities of k-hop neighborhoods is both intuitive and mathematically grounded. It provides a meaningful bridge between substructure-level and global structural information."*
- **The proposed method is recognized as novel and important within the graph transformer literature:**
	- `o64B`:  *"This formulation is both conceptually novel and technically elegant...The work makes a substantive contribution to the design of structure aware Graph Transformers."*
	- `o64B`:  *"GIST has clear potential to become a standard component in future graph Transformer architectures."*
	- `idmr`:  *"This paper introduces GIST (Gisty Intersection Signature Trait), a novel structural encoding method for Graph Transformers."*
	- `1jQu`:  *"...The reported results are pushing the boundary on a variety of standard benchmarks."*
	- `1jQu`:  *" This paper proposes its application also as a structural encoding for graph transformer architectures, which is a natural and promising next step."*
	- `1jQu`:  *"As stated in my review, I find the principal idea of using the techniques of Chamberlain et al. as a structural encoding promising."*

**We appreciate the recognition and couldn't ask for more for this type of work.**

---

> ### Author Response · Authors · 2025-12-01
> **Reviewers' Concerns and Our Rebuttals (Resolved)**
>
> We believe a summary is fair only if it includes both the strengths and the concerns raised by the reviewers. Therefore, we present below the main questions brought up during the rebuttal period, along with how we addressed each point, beyond minor content/motivation clarifications and presentation suggestions.
>
> Firstly, we provided a brief walkthrough of all the concerns raised during the review process and how we addressed them thoroughly. These points have been **explicitly acknowledged by nearly all reviewers who raised them (with the exception of R`o64B`, who do not have the opportunity to confirm)**:
> - **Clarification of technical presentation** (`fbzd`, `1jQu`):
>   We revised the notation, improved definitions, and reorganized explanations for clarity.
>   *Both reviewers confirmed that these issues were resolved.*
>
> - **Empirical analysis of how much performance gain comes from GIST features** (`o64B`, `idmr`):
>   We added controlled comparisons isolating the contribution of GIST’s structural features. *`idmr` acknowledged.*
>
> - **Ablation study on performance–efficiency tradeoffs** (`o64B`, `idmr`):
>   We added new evaluations covering hash parameters, estimator depth, and k-hop choices. *`idmr` acknowledged.*
>
>
> Additionally, reviewer `o64B` requested (i) pseudocode for GIST computation and (ii) analysis on the bias/variance of the estimation algorithm. Both items were added in the revision.

---

> ### Author Response · Authors · 2025-12-01
> **Reviewers' Concerns and Our Rebuttals (Pending)**
>
> Outside the resolved concerns mentioned above, we are left with **one remaining concern from reviewer `fbzd` and one from reviewer `1jQu`**.
>
> ### **Reviewer `fbzd` – “meaningful contribution / novelty”**
> - The only remaining issue is whether GIST makes a *“meaningful contribution”* beyond reusing MinHash and HyperLogLog.
> - This view **directly contrasts with the other three reviewers**, who state that:
>   - R `o64B`: *“This formulation is both conceptually novel and technically elegant.”*
>   - R `idmr`: *“This paper introduces GIST, a novel structural encoding method for Graph Transformers ... It provides a meaningful bridge between substructure-level and global structural information.”*
>   - R `1jQu`: *“This paper proposes its application also as a structural encoding for graph transformer architectures, which is a natural and promising next step.”*
> - We also note that reviewer `1jQu` initially also raised concerns related to novelty, but **after the first rebuttal** explicitly acknowledged that they *“find the principal idea of using the techniques of Chamberlain et al. as a structural encoding promising”*, and subsequently increased their overall recommendation.
> - Conceptually, GIST:
>   - Is **the first** to adopt intersection cardinality in Graph Transformers
>   - Is **the first** to **capture higher-order structural relationships at the graph level**, in contrast to Chamberlain et al., which targets **local subgraph sketches for the automorphic node problem**
> - Methodologically, the contribution is on par with, and in some respects even stronger than, widely accepted works like GRIT, which is considered impactful despite being built on long-established random-walk ideas.
> - Taken together, we believe the remaining disagreement reflects a stricter *personal novelty bar* rather than a genuine lack of meaningful contribution.
>
> ### **Reviewer `1jQu` — clearer positioning vs. Chamberlain et al.**
>
> Reviewer `1jQu` acknowledges that the **core idea is promising**, calling it *“a natural and promising next step”* and explicitly stating *“I find the principal idea … promising.”* Their only remaining concern is that the manuscript should **more clearly position GIST against Chamberlain et al.**
>
> We respectfully note that the reviewer’s remaining objection rests on a fundamental misunderstanding of Chamberlain et al.’s goal and scope:
>
> - Their method is explicitly framed around solving the **automorphic node problem** in GNNs via **local subgraph sketches**, not modeling global relational structure.
> - By contrast, GIST is designed to encode **graph-level** higher-order relationships across heterogeneous substructures in Graph Transformers—a different problem, with different claims and evaluation focus.
> - In other words, while we adopt the same estimators, the *question being answered* and the *structural quantity being modeled* are fundamentally different.
>
>
>
> We have revised the writing to make the positioning vs. Chamberlain et al. more explicit. Given these changes, and should there be sufficient time and space in a final version, we believe our rebuttal and revisions are enough to resolve this remaining, largely presentational concern about positioning, without affecting the substance or validity of the contribution.
>
> ---
>
> In essence, we would like to respectfully ask the AC:
>
> > ## **`Do these remaining concerns truly rise to the level of being deal-breaking, such that they should disqualify an otherwise strong and publishable work?`**
>
> We genuinely believe they do not—they are either matters of positioning, or differences in personal novelty thresholds that can be fully addressed with more space and iteration of rebuttal. That said, we fully trust and respect your judgment, and we will accept your decision regardless.
>
>
> Warm regards,
> *Paper13433* Authors

---

### Meta-Review · Area_Chair_uQTB · 2026-01-06

**Summary:**

This paper introduces GIST (Gisty Intersection Signature Trait), a structural encoding for Graph Transformers based on k-hop neighborhood intersection cardinalities between node pairs. The method demonstrates consistent performance improvements across several benchmark tasks, including ZINC, Peptides, and MoleculeNet.

The paper received mixed reviews, with two reviewers supporting acceptance and two leaning toward rejection. While I acknowledge the strong empirical performance of the proposed approach, after carefully reviewing the paper and the discussion, I share the concerns raised by reviewers ``fbzd`` and ``1jQu``. In particular, I am not convinced that the work constitutes a significant novel methodological contribution in light of Chamberlain et al. (2022). Extending their structural encodings to Transformer-based models for graph-level tasks appears rather incremental. The theoretical analysis is helpful but not particularly strong. Moreover, adequately positioning the contribution would require substantial text revision. For these reasons, I recommend rejection this time.

**Reviewer Concerns:**

Reviewers have raised some concerns, including limited novelty, unclear theoretical contributions, insufficient acknowledgment of prior work, lack of comparisons against other baselines, lack of experiments on large datasets, and missing ablation studies, among others.

Most concerns were addressed by authors with clarifications and additional experiments -- in particular, all concerns by reviewers ``o64B`` and ``idmr`` were addressed in my opinion. Nonetheless, the following concerns remain unresolved.

**Novelty**. Reviewers ``fbzd`` and ``1jQu`` raised important concerns regarding novelty wrt Chamberlein et al. (2022), noting that both works rely on the same structural features.

In their reply, authors have argued that Chamberlein et al. tackle automorphic node identification problem in link prediction while their work focus on capturing higher-order structural relationships between node pairs. The authors also assessed a simple baseline leveraging a vertex-level variant of structural encodings based on common neighbor features, showing it performs worse than the proposed method. Beyond this, the authors attempt to argue novelty by contrasting their approach with other accepted papers, which I find unconvincing.

I share the novelty concerns raised by ``fbzd``. In particular, Chamberlain et al. employ the exact same (pairwise) descriptors as edge features in GNNs for link prediction tasks. First, the close relationship between GNNs and self-attention mechanisms in Transformers is well established [1]. Leveraging structural features to bias attention scores in Transformers is therefore methodologically very similar to using them to modulate message-passing aggregation in GNNs. In addition, adapting GNN architectures from link prediction to graph-level tasks typically only requires modifying the readout layer. Overall, I believe the authors did not convincingly articulate meaningful design differences relative to Chamberlain et al.

**Significance of theoretical contributions.** Reviewers ``fbzd`` and ``1jQu`` also questioned the significance of the theoretical contributions.

While the theoretical results provide some support for the proposed method, I agree with the reviewers that they are relatively weak. In expressivity analysis, results showing incomparability between model classes (i.e., neither is strictly more expressive than the other), or demonstrating that one class is not less expressive than another via a single counterexample, are typically considered the weakest forms of theoretical guarantees. This is the case for statements 2 and 3 in Theorem 3.3. Furthermore, the analysis seems to focus on the expressivity of GIST alone, rather than the full model (GIST combined with Transformers), which is ultimately used in practice. I believe it would be interesting to study the expressive power of the combined models (structural encodings + Transformers/GNNs).

Additionally, although expressivity is used as a motivating argument throughout the paper, the method is not evaluated on expressivity benchmarks [2]. Finally, reviewer ``1jQu`` explicitly asked the authors to position their method within the Weisfeiler–Lehman hierarchy (e.g., relative to 2-FWL), which was not addressed in the rebuttal.

[1] C. Joshi. Transformers are Graph Neural Networks, Arxiv 2025.

[2] An Empirical Study of Realized GNN Expressiveness, ICML 2024.

**Insufficient positioning**. Despite revisions at L182, L228-229, and L501-508, reviewer ``1jQu`` maintains:

- “even in the revision, presents the idea as if conceived by the authors, with an unassuming reference to Chamberlain et al."
- “first framing things slightly different on the surface (intersection signatures vs. subgraph sketching) before falling back on precisely the methods of prior work when it comes to practical implementation."

The authors argue they don’t use "subgraph sketching" terminology because they encode "the entire graph structure and higher-order relationships" rather than local subgraphs. However, this distinction appears largely semantic: both approaches compute intersection cardinalities of k-hop neighborhoods using essentially identical techniques.

**Reviewer Scores:**

**Reviewer ``fbzd``** acknowledged the rebuttal and increased score from 2 to 4 while maintaining reservations:

> "Although I am still not fully convinced that is achieved in this work given the updated limited scope on the theoretical side, the additional experiments with GIST-PE are an encouraging start in that direction."

Based on this, I believe the reviewer would likely have maintained this score even if the discussion had continued.

**Reviewer ``1jQu``** also increased score (possibly from 2 to 4) but emphasized:

> "I wish to highlight to the AC that my main concerns are misrepresented by this final rebuttal comment in the chain. My issue was not with the theoretical foundation but with (1) problematic/false overclaiming regarding the expressivity results, and with opaque technical presentation (this has been resolved), and (2) with, in my eyes, an inappropriate presentation of the scientific context that this work emerged from."

In my opinion, the authors’ response did not fundamentally address the novelty and positioning concerns raised by the reviewer in a way that would justify leaning toward acceptance.

**Reviewer ``idmr``** maintained positive score (6) after confirming concerns were addressed.

**Reviewer ``o64B``** provided strong initial support (8) but didn’t participate in the discussion.

---

### Decision · Program_Chairs · 2026-01-26

Reject